# Slow-to-fast transition of giant creeping rockslides modulated by undrained loading in basal shear zones

Federico Agliardi [1✉], Marco M. Scuderi [2], Nicoletta Fusi[1] & Cristiano Collettini[2,3]

Giant rockslides are widespread and sensitive to hydrological forcing, especially in climate change scenarios. They creep slowly for centuries and then can fail catastrophically posing major threats to society. However, the mechanisms regulating the slow-to-fast transition toward their catastrophic collapse remain elusive. We couple laboratory experiments on natural rockslide shear zone material and in situ observations to provide a scale-independent demonstration that short-term pore fluid pressure variations originate a full spectrum of creep styles, modulated by slip-induced undrained conditions. Shear zones respond to pore pressure increments by impulsive acceleration and dilatancy, causing spontaneous deceleration followed by sustained steady-rate creep. Increasing pore pressure results in high creep rates and eventual collapse. Laboratory experiments quantitatively capture the in situ behavior of giant rockslides and lay physically-based foundations to understand the collapse of giant rockslides.

[1] Department of Earth and Environmental Sciences, University of Milano-Bicocca, Piazza della Scienza 4, 20126 Milano, Italy. [2] Department of Earth Sciences, La Sapienza University of Rome, Piazzale Aldo Moro 5, 00185 Roma, Italy. [3] Istituto Nazionale di Geofisica e Vulcanologia (INGV), Via di Vigna Murata 605, 00143 Roma, Italy. ✉email: federico.agliardi@unimib.it

Giant creeping rockslides in crystalline rocks represent major threats to human life and infrastructures[1,2] due to their volume ($10^6$–$10^8$ m$^3$) and fragmentation potential, resulting in extremely high kinetic energy mobilization if massive collapse occurs[3,4]. However, these landslides evolve over thousands of years and establishing reliable criteria to predict their critical transition to fast movements and catastrophic failure remains challenging[5–7].

Large natural slopes are sub-critically stressed (i.e., subjected to stress condition slightly lower than their instantaneous strength) and respond to major geomorphic perturbations (e.g., river incision in uplifting settings, glaciation/deglaciation) by progressive rock failure processes[8,9]. Over time, these processes lead to rock damage accumulation and permeability enhancement, until strain localizes along basal shear zones similar to tectonic faults[10–12]. As rockslide shear zones accumulate strain, they become thicker and richer in fine-grained gouge, causing permeability reduction and favoring the development of perched aquifers systems[9,13–16] (Fig. 1). Progressive rock failure and damage accumulation along basal shear zones result in commonly observed slope creep, that becomes dominated by the hydro-mechanical behavior of the shear zone as it evolves toward maturity[14–16].

While giant mature rockslides creep slowly at steady rates for years under drained hydraulic conditions[6,17,18], pore fluid pressure perturbations can modify the stress state within the landslide, resulting in sudden or delayed acceleration pulses, long periods of sustained steady displacement rates, or runaway rupture resulting in catastrophic collapse[7,16–21]. Studies based on remote sensing and simplified modeling showed that slow, creeping landslides are widespread and particularly sensitive to dramatic short-term changes of hydrological boundary conditions, especially in climate change scenarios[22,23]. Capturing the coupling between hydrological and mechanical processes at the origin of diverse creep styles of giant rockslides is key to predict their slow-to fast transition. Nevertheless, establishing physically-based criteria useful to predict the catastrophic evolution to failure of giant creeping rockslides remains a major challenge.

The relative importance and interplay of the basic physical mechanisms underlying these processes on the in situ scale of large rockslides remain poorly understood. As a result, our knowledge of landslide time-dependent motion and transient response to hydrologic triggers relies on the statistical analysis of rainfall, groundwater and displacement time series[7,19,20,24,25] and on mathematical models based on simplified viscous rheology[6,16,26,27]. Although useful and relatively easy to apply, these approaches are unable to account for the full spectrum of rockslide slip behaviors, and leave considerable uncertainties that impact the reliability of forecasting models and early warning criteria[28].

Some authors applied rate- and state-dependent friction models to describe landslide motion accounting for both slow-sliding and catastrophic collapse[29,30]. Others invoked undrained hydro-mechanical coupling and dilatant strengthening mechanisms to explain episodic and time-variable patterns of landslide motions[18,31–33]. However, the application of these models to giant slow-moving rockslides lacks experimental support and a direct comparison between in situ and laboratory data. Moreover, laboratory studies on landslide dynamics mainly focused on the residual strength properties of shear zones[12,34–38] and the evolution of undrained failure mechanisms in shallow flowslides[39,40]. As a result, there is a paucity of rate-and-state friction parameters measured under landslide stress conditions and for adequate rock types, forcing the formulations of models to be based on parameters derived from the fault mechanics literature[30].

Here we address this problem by integrating innovative laboratory experiments on cataclastic materials, sampled from the basal shear zone of a giant creeping rockslide, with in situ monitoring observations. We use these data to explore the rockslide slip behaviors resulting from the interplay between pore-pressure perturbations and rate- and state-dependent[41] frictional properties of the shear zone. We demonstrate that the spectrum of rockslide creep styles until catastrophic collapse is regulated by undrained hydro-mechanical response to short-term fluid pressure perturbations.

## Results

**The Spriana rockslide.** The Spriana rockslide in Val Malenco (Italian Central Alps; Fig. 1a) affects a 1200 m high slope that was carved by fluvial and glacial erosion in a compact granodioritic gneiss[42,43]. The rockslide nucleated in Early Holocene and evolved during the last 6 kyr[9] until early 20th century, when it started a fast evolution with major acceleration periods in 1960 and 1978. These accelerations caused major geomorphological modifications, evacuation of two villages and a long-term threat pending on the town of Sondrio[44,45]. The area was deeply investigated by geological and geophysical surveys, 20 full-core deep boreholes and deformation monitoring, allowing the reconstruction of rockslide anatomy[9] (Fig. 1b).

The rockslide affects 50 Mm$^3$ of rock material with a basal shear zone up to 95 m deep, made of cataclastic granular material up to some meters thick and connected to three main scarps reaching the surface at different elevations[9]. A thin aquifer is suspended on the basal shear zone[44,45]. Surface displacement monitoring, performed through topographic and ground-based measurement techniques (wire extensometers), shows that the rockslide creeps at steady long-term rate of 0.5–3 cm yr$^{-1}$, while episodic or prolonged acceleration periods, followed by decelerations, are superimposed on the long-term creep rate (Fig. 1c, d). Rockslide acceleration pulses are associated to the recharge of the perched groundwater table following intense or prolonged rainfalls, with major activation periods occurring when the water table rise exceeds 2 m (Fig. 1c, d). The analysis of subsurface investigation and monitoring data, including high-quality drill-cores, borehole inclinometer measurements, topographic and extensometer surface displacements data, suggest that the rockslide mass is affected by relatively small internal deformation[9,44,45], so that long-term surface displacements are mainly related to hydro-mechanical forcing along the basal shear zone.

**Shear zone hydro-mechanical properties.** The basal shear zone of the rockslide (Fig. 1b) is made of angular clasts up to 2.5 cm in a silty-sandy matrix, with average in situ bulk density of 1.92 g cm$^{-3}$ and a cataclastic texture like that of fault rocks[46]. Gouge material (<0.6 mm) is a mixture of quartz, K-feldspar, amphibole, chlorite and white mica, with a total phyllosilicate content of about 40% and a mineral composition poorly varying in different grain size fractions (Supplementary Fig. 1). The measured steady-state friction coefficient of the water-saturated gouge is $\mu = 0.51$ (Fig. 2a and Supplementary Fig. 2), equivalent to a friction angle $\phi = 27°$. This is poorly affected by grain size, close to the values obtained by the back-analysis of global slope stability ($0.5 < \mu < 0.53$; $26.6° < \phi < 27.9°$)[44], and consistent with previous laboratory measurements on rockslide shear zones with comparable mineralogy ($0.51 < \mu < 0.52$; $27° < \phi < 27.5°$)[12]. The frictional stability of the shear zone material is evaluated within the framework of the rate- and state-dependent constitutive law, by performing velocity-step experiments over the velocity range of slow-to fast transition of giant rockslides, 1–300 µm s$^{-1}$ (see Methods). Results show that the quartz-phyllosilicate gouge has a rate-strengthening behavior, more prone to slow creep than to

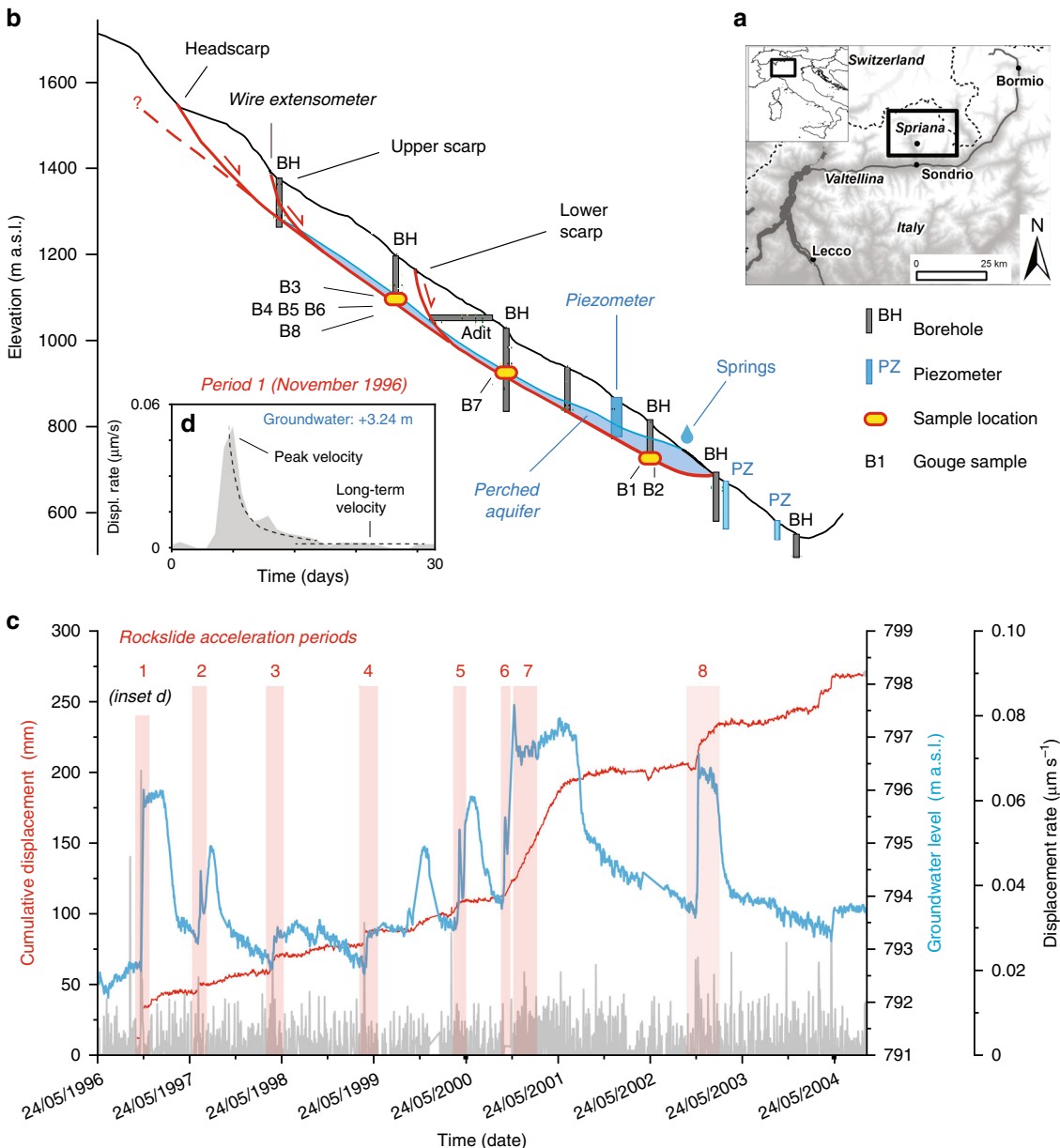

**Fig. 1 Spriana rockslide. a** Location map; **b** geological and hydrogeological model, with basal shear zone sampling locations, sample numbers and location of field measuring devices used for the analysis. The representation of the perched aquifer integrates borehole and piezometer observations made upon 1989–90 site investigations. Horizontal and vertical distance are same scaling; **c** monitoring time series of rockslide surface displacement (wire extensometer) and groundwater level (hydraulic borehole piezometer), with selected rockslide acceleration periods outlined; **d** detail of acceleration impulse associated to a period of increase of groundwater level (acceleration period n.1) and characteristic descriptors: time to peak, peak velocity, long-term velocity.

catastrophic collapse, fading to rate-neutral when water-saturated (Fig. 2b). The saturated hydraulic conductivity of the gouge, that was measured under in situ stress conditions ($4 \times 10^{-10}$ m s$^{-1}$ at effective normal stress of 2 MPa; Fig. 2a), is consistent with previous data available for rockslides with similar mineralogy[17].

**Shear zone response to stress and pore-pressure changes**. We explored the hydro-mechanical processes regulating the time-dependent slip behavior of the rockslide by performing non-conventional creep experiments. In these experiments, shear stress is maintained constant at a specified subcritical value (i.e., a fraction of the instantaneous shear strength, $\tau_s$) and pore pressure is increased stepwise, resulting in a systematic reduction of the effective normal stress (Figs. 2c and 3). This technique allows

reproducing the long-term loading conditions of the Spriana rockslide and their modifications when the short-term rise of water table (i.e., increase in pore pressure) promotes the onset of episodic or prolonged acceleration periods (Fig. 1c, d).

During the pore pressure-step creep experiments, the short-term increase of pore pressure at constant shear stress results in a complex slip behavior (Fig. 3a, b) that is persistent for different levels of applied shear stress (Supplementary Figs. 3 and 4). Each pore pressure increase causes a sudden slip acceleration accompanied by shear zone dilation (Fig. 3c). The attainment of peak velocity is systematically delayed in respect to the pore pressure step (Fig. 3d–f) and the delay becomes shorter with increasing pore pressure, as the stress state approaches the Mohr-Coulomb failure envelope (Figs. 2c and 3a). Each acceleration

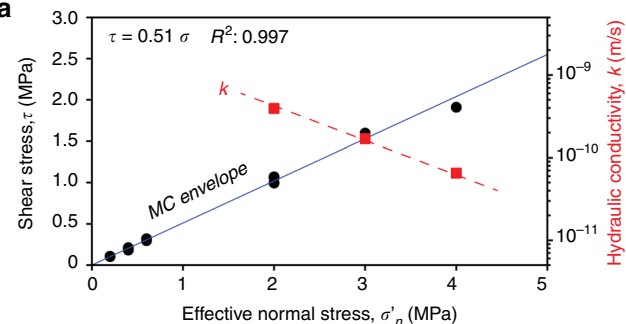

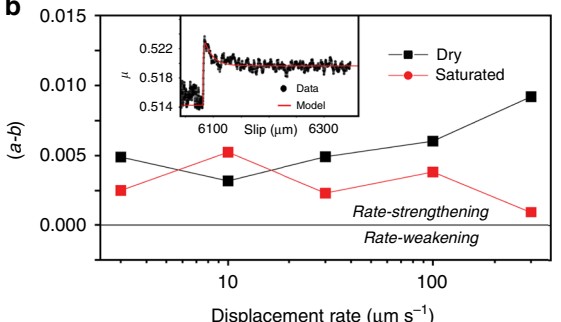

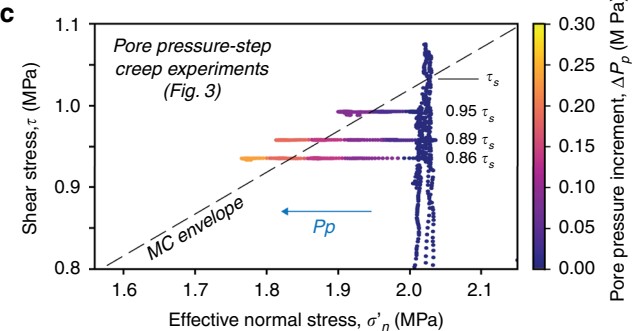

**Fig. 2 Friction, fluid flow properties, and stress evolution during fluid pressurization. a** Frictional and hydraulic properties of the tested natural gouge, derived from stable-sliding laboratory tests; **b** rate-and-state frictional properties of the tested natural gouge, derived from velocity-step tests. Inset shows a typical velocity step along with the model inversion performed to retrieve the $(a–b)$ values; **c** experimental stress paths during pore-pressure-step creep experiments. In these experiments, we started with an initial stage at constant displacement rate ($10 \, \mu m \, s^{-1}$) to achieve the instantaneous shear strength ($\tau_s$). Then we fixed constant shear stress at 86, 89, and 94% of $\tau_s$ and started creep experiments (details are reported in method and Fig. 7). Errors (standard deviation) are smaller than symbols size.

pulse is followed by a self-deceleration stage, Creep I, evolving in a long-term steady slip rate of finite magnitude, Creep II (Fig. 3b), up to ten times lower than the peak velocity. Shear zone deformation in Creep II is associated to gouge compaction that increases with increasing slip rate (Figs. 3c and 4). The self-decelerating style of shear zone deformation becomes less effective as pore pressure rises, resulting in increasing values of Creep II slip rates (Fig. 3b). As the stress state approaches the Mohr-Coulomb failure envelope (Fig. 2a), a pore-pressure step results in an initial peak velocity followed by a short-term velocity reduction that eventually evolves into accelerated creep (Creep III) and catastrophic failure (yellow paths in Fig. 3a–c). Experiments performed at different shear stress levels (Fig. 2c) mimic rockslide basal shear zone inclinations or changes of

external loads. These experiments show that for higher shear stresses the peak slip velocity is attained in a short time and few or small pore pressure increments can trigger the transition to accelerated creep III quickly or in a nearly instantaneous fashion (Fig. 3, Supplementary Figs. 3 and 4).

## Discussion

During the pore pressure-step creep experiments, the system consistently behaves as described above when measured slip rates overcome a certain velocity threshold (Fig. 4). Above this threshold, the systematic observation of shear zone dilatancy corresponding to short-term pore-pressure changes suggests the occurrence of direct, undrained fluid-to-solid hydro-mechanical coupling[47,48].

When a pore pressure step perturbs the internal stress state of the shear zone, simulating groundwater recharge by rainfall, rapid snowmelt or earthquake shaking, the shear zone shows complex slip behavior, unveiling the intimate coupling between hydrological and frictional processes. As suggested by the theoretical model of Iverson[33] and supported by our observations, the system response to fluid pressurization initially shows a short-term slip acceleration, followed by undrained dilation causing subsequent deceleration. Increasing pore pressure in the shear zone reduces the effective stresses and triggers accelerated slip causing undrained response by shear zone dilatancy[47] for slip rates larger than $0.05 \, \mu m \, s^{-1}$ (Fig. 4). The resulting negative pore-pressure feedback causes dilatant strengthening and a spontaneous deceleration[48], countering the effects of frictional weakening caused by pore pressure increase.

Following the self-deceleration stage, slip continues at a sustained long-term steady rate (Creep II), due to the evolution of the loading conditions from undrained to semi-drained, promoted by gouge fabric rearrangement and fluid diffusion[49,50] and the rate-strengthening/neutral slip behavior of the shear zone (Fig. 2b). Creep II occurs via shear zone compaction that is directly proportional to slip rate (Figs. 3a–c and 4). For higher shear stress and pore-pressure increase, the effects of dilatant strengthening are faster but less effective, resulting in a higher Creep II slip rates (Fig. 4). This mechanism remains operational until the increase in pore pressure reduces the effective stress to values corresponding to the critical stress state for failure (Fig. 2c), causing the onset of accelerated creep (Creep III) and catastrophic failure (Figs. 3a and 4). In this view, secondary creep can be viewed as a condition of dynamic equilibrium between the shear zone weakening induced by pore pressure increase, the dilatant strengthening promoted by undrained hydro-mechanical coupling, and the velocity strengthening/neutral behavior, with the latter tempering the rockslide attitude to catastrophic collapse.

The number of pore pressure steps required to reach critical failure conditions depends on the initially mobilized shear strength of the gouge material (Fig. 2c) and on the magnitude of individual applied pore-pressure steps (Fig. 4). High shear stress values (representing steep or loaded slopes) or large pore-pressure steps (mimicking the effects of extremely intense rainfall) can result in ineffective dilatant strengthening and nearly instantaneous failure.

The proposed mechanism can act in concert with others, as proposed in the literature, to explain the observed spectrum of slip behaviors of landslides on the grounds of rate-and-state frictional models[29] and also accounting for the force balance between the sliding surface and the surrounding loading medium[30]. However, for giant rockslides where episodic or prolonged acceleration periods are mainly forced by pore-pressure builds-up, our data indicate that hydro-mechanical coupling related to short-term pore-pressure development is the major driver of time-dependent slip behavior.

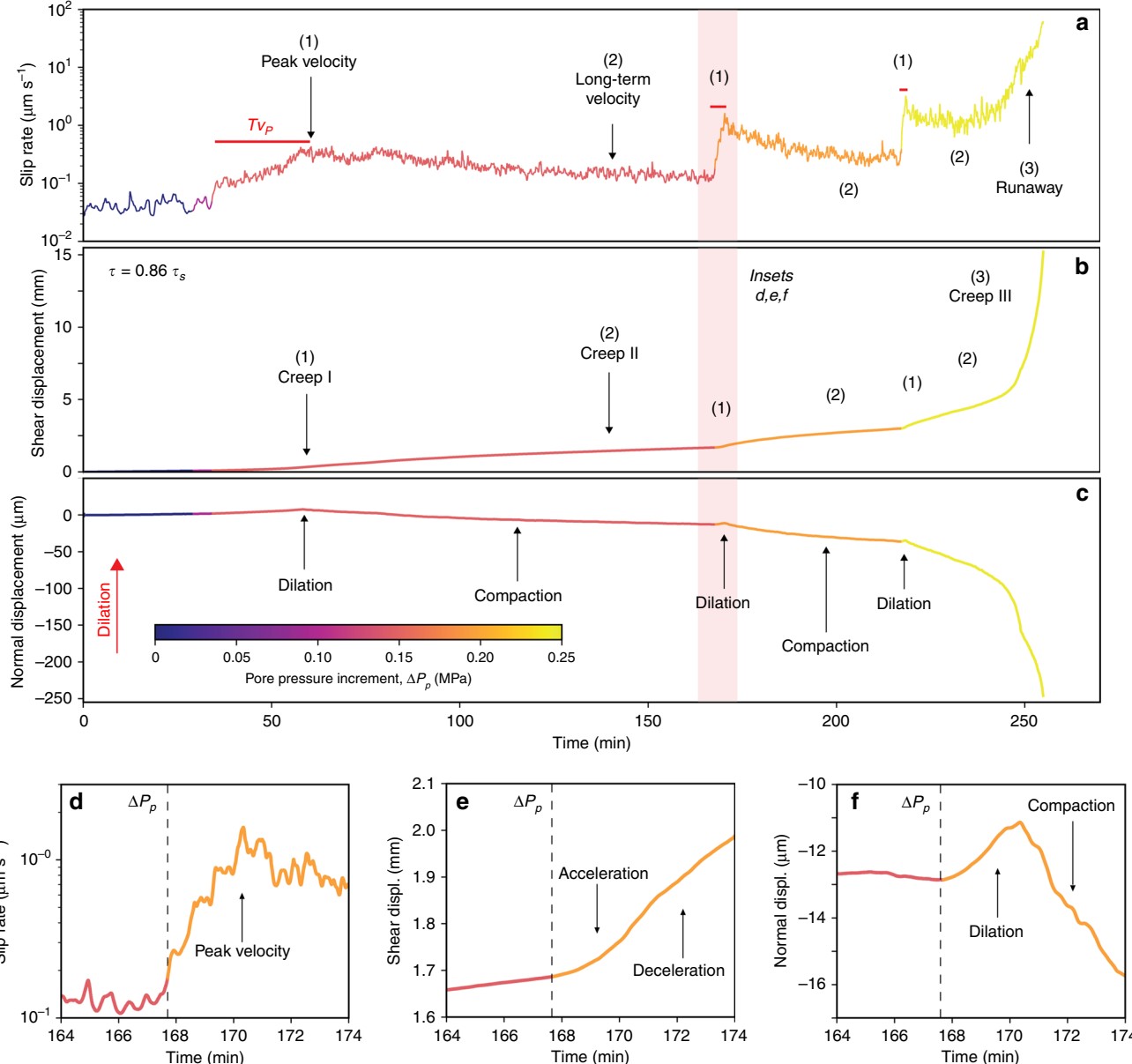

**Fig. 3 Shear zone evolution during laboratory pore pressure-step creep experiment ($\tau = 0.86\tau_s$).** Plots of **a** slip (shear displacement) rate, **b** cumulative shear displacement, and **c** normal displacement versus experiment time. The red bars in the first panel of the figure mark the time required to reach peak slip velocity ($TV_P$) following a pore-pressure step. The numbers highlight: (1) peak velocity followed by the onset of Creep I, (2) long-term steady slip, i.e., Creep II; (3) accelerated creep, i.e., Creep III; **d**–**f** details of one pressure step (pink shaded area in (**a**–**c**)). The peak slip velocity is attained after some minutes from the pore-pressure step and then the shear zone decelerates. Acceleration is accompanied by shear zone dilation whereas deceleration results in compaction, illustrating the typical undrained response of the shear zone to a step of pore-pressure increase.

A rockslide deformation style, characterized by sudden acceleration and self-deceleration following short-term events of pore pressure increase, has been observed both in situ (Fig. 1c and Supplementary Table 1) and during laboratory experiments (Fig. 3). To compare the two datasets quantitatively, we used the following descriptors (Fig. 5a and Methods) quantified from in situ (Fig. 1c, d) and laboratory observations (Fig. 3, Supplementary Figs. 3 and 4): (a) time between pore pressure stepwise increment and the velocity peak ($TV_P$); (b) peak velocity ($V_P$) associated to the acceleration impulse occurring after each pore-pressure increase; (c) long-term velocity ($V_{CII}$) in the Creep II stage. The comparison of in situ and experimental rockslide behavior (Fig. 5b–d) demonstrates that the hydro-mechanical response of the rockslide to short-

term pore pressure variations at the shear zone level is quantitatively consistent with laboratory results. We can observe this consistency despite the spatial scale and complexity of the natural system, that includes thick shear zones with variable textural maturity and is affected by brittle deformation within the moving rockslide mass.

Laboratory values of time to peak velocity (normalized $T_{VP}$; Fig. 5b) decrease when pore pressure and mobilized shear strength increase (Fig. 5b; statistics in Supplementary Table 2), while experimental values of peak velocity ($V_P$, Fig. 5c) show strong positive correlations with pore-pressure increase (Spearman's $\rho$ reaching 0.8–1; Supplementary Table 2). Rockslide in situ values of both $T_V$ and $V_P$ fall within the experimental bounds and follow the same general trends but are noisier and weakly

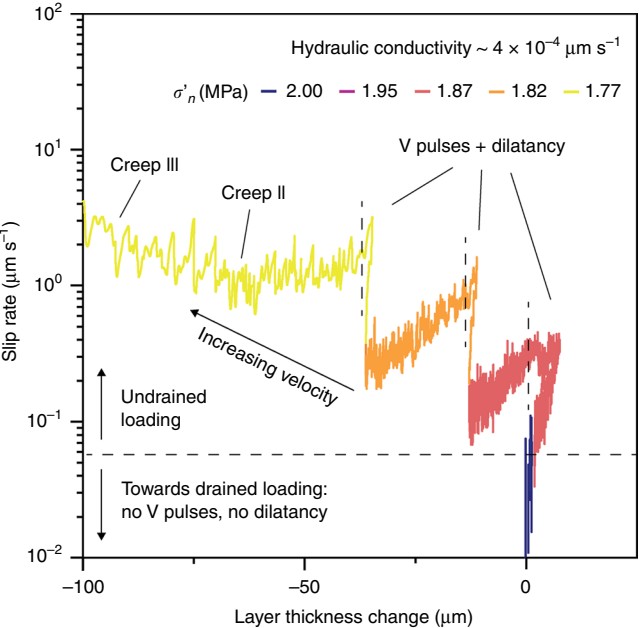

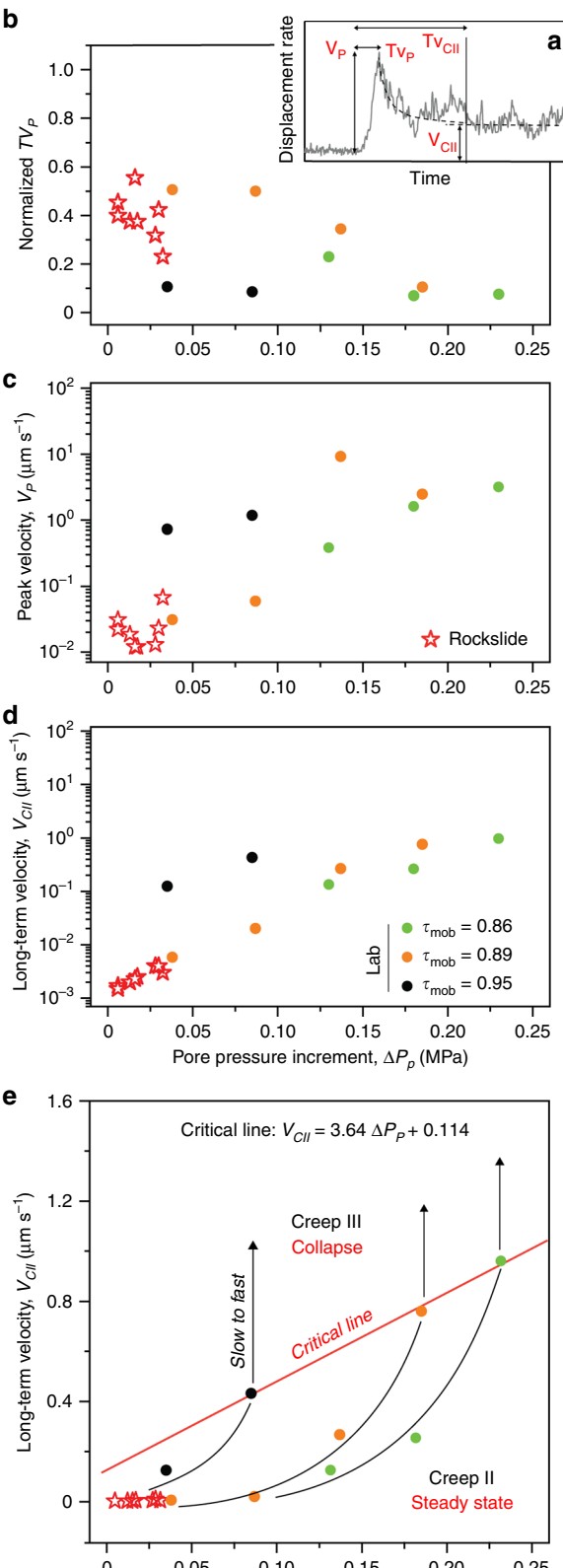

**Fig. 4 Evolution of dilatancy/compaction, slip rate, and creep style during pore-pressure steps.** Applied shear stress: $0.86\tau_s$ (see Fig. 3). The different colors refer to different values in the effective normal stress as reported in the caption. When the slip rate along the experimental shear zone exceeds a threshold (here $0.05\ \mu m\ s^{-1}$), loading conditions become markedly undrained. For these conditions, each pore pressure step is associated to acceleration and dilation followed by compaction and the attainment of a new long-term Creep II-velocity ($V_{CII}$): note that $V_{CII}$ increases with increasing pore pressure.

correlated with pore pressure increase. It suggests that, for a given level of mobilized strength, the short-term hydro-mechanical response of both laboratory and in situ systems becomes systematically faster and more intense as pore pressure increases, although in situ systems show noisy behaviors over short periods (days to weeks; Supplementary Table 1). In contrast, the steady Creep II-velocity ($V_{CII}$, Fig. 5d) shows a strong log-linear correlation with pore pressure increase, and this strong correlation yields for both laboratory and rockslide in situ data (Pearson's $r$ coefficient always exceeding 0.9 and close to statistical significance; Supplementary Table 2).

The close and statistically significant similarity between the values of $V_{CII}$ observed in situ and in the laboratory at mobilized shear strength of $0.89\tau_s$ ($t$-value: $-1.483$; Prob> $|t| = 0.19$) suggests that about 90% of the instantaneous strength of the rockslide basal shear zone is mobilized. Furthermore, the observation that the rockslide has never reached tertiary creep is consistent with the low pore-pressure increments observed by field groundwater monitoring (Fig. 1c). In fact, the maximum observed short-term increase in groundwater level measured in the observation period reached 3.24 m, resulting in a pore pressure increment of 0.032 MPa, while, to reach critical conditions it should reach about 19 m, i.e., pore pressure increment of 0.185 MPa. This high increase in groundwater level is consistent with the values observed in alpine rockslides of similar magnitude during periods of critical acceleration[6]. The quantitative agreement between laboratory and in situ observations supports the hypothesis that the rockslide motion is regulated by undrained hydro-mechanical processes, occurring in the water-saturated basal shear zone, that are more effectively captured in the laboratory.

**Fig. 5 Laboratory vs in situ undrained response to short-term pore-pressure increase. a** Key descriptors for the system response to pore-pressure increments, $\Delta P_p$; **b** normalized time to reach peak velocity ($TV_P$); (**c**) peak velocity ($V_P$); **d** long-term Creep II-velocity ($V_{CII}$); **e** description of the rockslide path towards catastrophic collapse and threshold conditions (Critical Line; linear fit $R^2$: 0.992, $p$-value: 0.057) as a function of $V_{CII}$ and pore-pressure increment.

Our experiments on real rockslide shear zone materials show the full spectrum of creep behaviors commonly observed in giant mature rockslides in crystalline rock[6,7,16]. In particular, they reproduce the in situ response of the Spriana rockslide shear zone to pore-pressure fluctuations without significant scale effects, and shed light on the hydro-mechanical interactions that underlie different stages of slope creep and modulate the slow-to-fast transition to catastrophic failure (Fig. 5e).

Our results provide new insights towards an improved understanding of the hydro-mechanical coupling at the origin of rockslide movements, with the potential to advance capabilities to prevent rockslide disasters. In fact, impulsive slip rates recorded by in situ rockslide monitoring systems after intense rainfall or snowmelt are often high, leading to civil protection alarms but later recovered without a catastrophic follow-up[7]. Nevertheless, long-term, steady slip rates can remain high for months and empirical/statistical forecasting approaches[17,24,25] are usually unable to predict their possible transition to tertiary creep.

Our results suggest that empirical failure forecasting approaches can be poorly effective with giant rockslides because they are not able to separate the effects of short-term and long-term hydro-mechanical processes promoting and tempering the evolution of displacement towards a catastrophic outcome. Our data show that the descriptors of short-term hydro-mechanical response of rockslides to fast pore pressure increase are predictable in the laboratory but noisy in situ, hampering the establishment of simple empirical relationships between groundwater levels and rockslide displacements and rates[6,7,18,28]. In contrast, statistically robust correlations between the long-term velocity ($V_{CII}$) and pore pressure increase in specific scenarios of shear strength mobilization may provide a robust descriptor to establish empirical threshold conditions for transition to giant rockslide collapse (Fig. 5e). The rates of system response predicted by our experiments, that are quantitatively consistent with in situ rockslide behavior, scale log-linearly with pore pressure increment, providing a statistically significant description of the rockslide path toward catastrophic failure that is unaffected by short-term hydro-mechanical effects (Fig. 5e). Values of long-term velocity that correspond to a slow-to-fast transition to Creep III and runaway failure are well fitted by a simple linear envelope (Fig. 5e). These data provide physics-based, scale-independent foundations to improve forecasting models and hence prevent catastrophes related to giant mature rockslides in crystalline rocks, particularly those that still have not reached tertiary creep conditions.

## Methods

**Sample characterization and preparation.** We collected eight samples of cataclastic granular material from high-quality cores drilled across the rockslide basal shear zone in 1989 (Figs. 1, 5 and Supplementary Fig. 1); details of borehole investigations in refs. [9,44]:

B1 and B2 were sampled at 66.8–69.2 m in depth near the rockslide toe, close to the location of the groundwater monitoring point considered in our study;
B7 came from the 78.20–79.2 m depth interval, downslope of the rockslide lower scarp;
B3, B4, B5, B6, and B8 were sampled between 87.50 and 94.10 m in depth, just upslope of the rockslide lower scarp where the basal shear zone is more localized[9].

Core segments were scanned using MicroCT (BIR Actis 130/150; UNIMIB Rock Mechanics laboratory) for texture reconstruction and in situ bulk density estimation. Samples were scanned at 500 μm voxel size and reconstructed using ImageJ[51] and Avizo Fire (FEI-VSG™). Then, core samples were cut longitudinally, and one half was archived (Fig. 6).

We selected samples B2, B3, B4, B5, B7, B8 for grain size distribution and mineral composition analysis. Grain size distribution was determined by wet sieving[52], using 4.75, 2, 1.18, and 0.6 mm sieves, and laser diffraction particle sizing of material finer than 1.18 mm, using a Malvern Mastersizer 2000E (particle size measuring range: 0.1–1180 μm). Sample grain size distributions were plotted as cumulative frequency curves (Supplementary Fig. 1), that allow classifying the

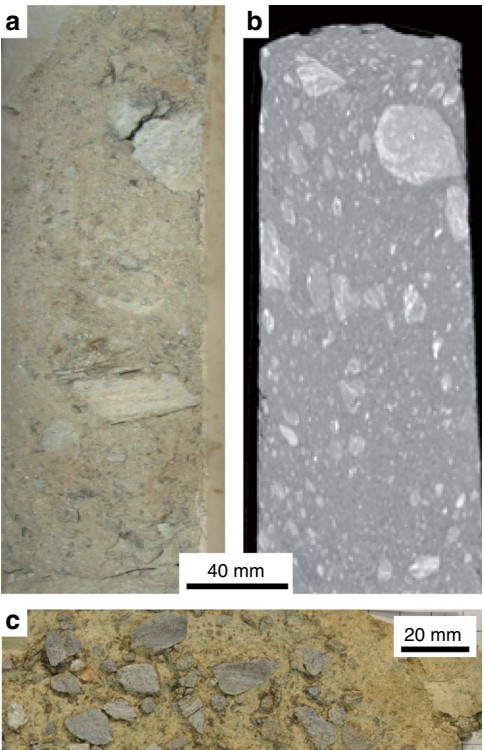

**Fig. 6 Example of cataclastic shear zone sample.** Sample B4 collected from the cores drilled in the Spriana rockslide: **a** core sample photo; **b** CT-scan (longitudinal slice); **c** and sample longitudinal cut.

material according to criteria commonly adopted for fault rocks[46] and quantifying the effective grain size $D_{10}$, i.e., 10th percentile of the grain size cumulative frequency distribution, correlated with permeability[52,53]. To identify possible compositional biases in the mechanical characterization of the sample material, we characterized the composition of different sample grain size fractions <2, <1, and <0.6 mm, respectively. Mineral composition was determined by XRPD, using a PANalytical X'Pert PRO diffractometer (2theta range: 3–80°, step size: 0.0167°) and performing quantitative analysis with GSAS[54]. The abundance (Wt%) of different mineral species (quartz, Qz; white mica, Wm; k-feldspar, Fls; chlorite, Chl; amphiboles, Amph) in different samples and "gouge scenarios" was portrayed by box-plots[55] (Supplementary Fig. 1).

Sample B4 (Supplementary Fig. 1), i.e., the richest in fine matrix, was selected for laboratory experiments. For the three gouge scenarios of B4, we characterized frictional strength using a conventional direct shear box (Wykeham Farrance Shearmatic; UNIMIB Soil Mechanics Laboratory) in consolidated-drained conditions[56]. For each gouge scenario material, three direct shear tests were performed at effective normal stress ($\sigma'_n$) of 0.2, 0.4, and 0.6 MPa, respectively (Supplementary Fig. 2). Mohr-Coulomb residual failure envelopes were obtained as linear fits of failure stresses after six shearing cycles (i.e., shear box runs). Best-fitting linear functions and 95% confidence intervals were obtained using the software OriginPro9™ (OriginLab Corp.) with a null cohesive strength constraint (Supplementary Fig. 2).

Specimens for further laboratory experiments within the BRAVA apparatus were prepared from the grain size fraction <0.6 mm of sample B4, representative of the natural composition of the rockslide basal shear zone gouge and its bulk permeability.

**Experimental set-up and procedures.** We performed laboratory experiments using a biaxial apparatus (BRAVA, Brittle Rock deformAtion Versatile Apparatus) in a double direct shear (DDS) configuration within a pressure vessel[57,58] (Fig. 7). Normal and shear stress are applied via fast-acting hydraulic servo-controlled rams and measured within the pressure vessel using strain gauged hollowed load cells placed at ram nose (LEANE International mod. CCDG-0.1-100-SPEC), with an accuracy of ±0.03 kN over a maximum force of 1.5 MN. Displacements are measured via Linear Variable Displacement Transducers (LVDT) with an accuracy of ±0.1 μm (Fig. 7a). Load point displacement measurements are corrected for the stiffness of the apparatus, with nominal values of 386.12 kN mm⁻¹ for the vertical frame and 329.5 kN mm⁻¹ for the horizontal one.

The DDS configuration consist in a three-steel block assembly that sandwiches two identical layers of granular material (grain size <600 μm). Forcing blocks are

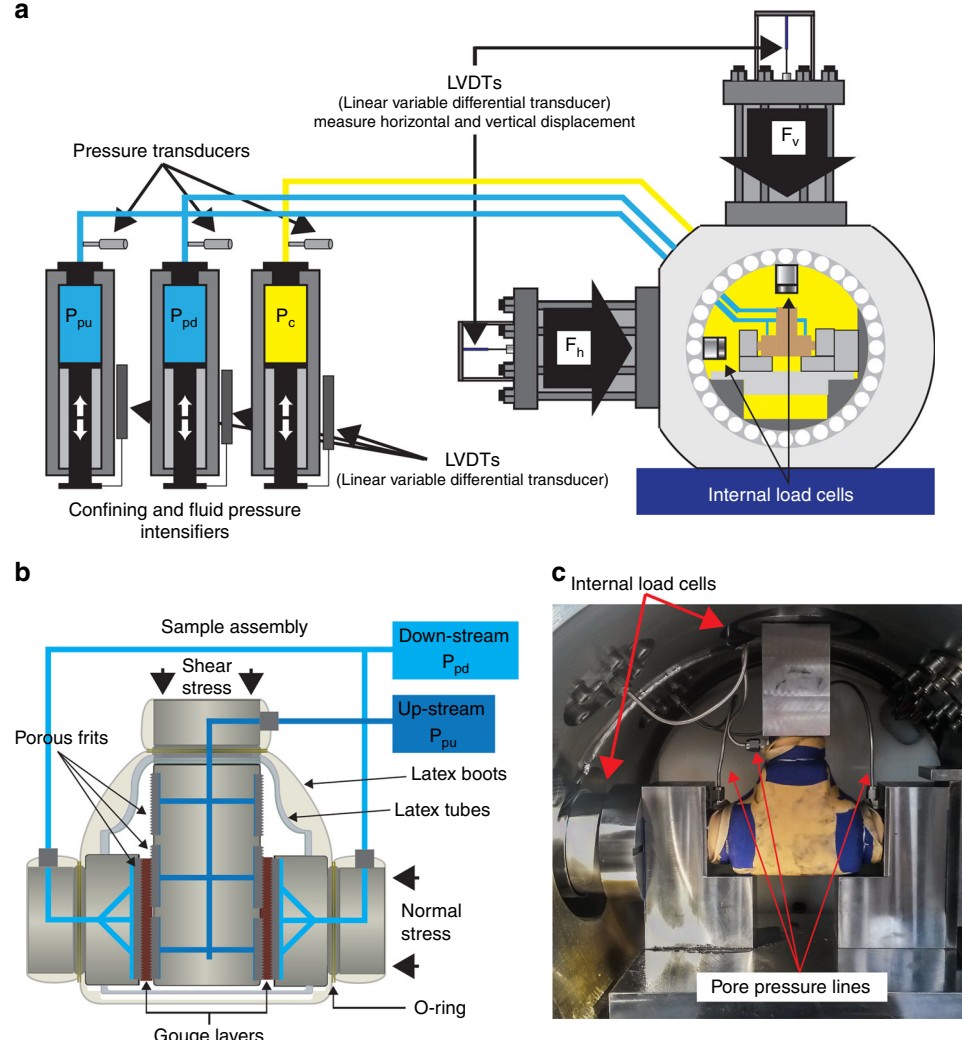

**Fig. 7 Experimental apparatus: BRAVA. a** Apparatus set-up, including the vessel, loading and measuring systems, and fluid pressure intensifiers; **b** sample assembly for biaxial shear experiments; **c** internal view of the vessel with sample assembled for experiments with confining pressure.

equipped with conduits for fluid flow that directly connect the sample layer with the up- and down-stream fluid pressure intensifiers (Fig. 7b). Sintered porous frits (permeability $1 \times 10^{-14}$ m$^2$) are press fit within cavities in the forcing blocks to allow homogenous distribution of fluids on sample surfaces. For this configuration, the nominal frictional contact area is $5.54 \times 5.55$ cm and we refer all the measurements of stress, displacement and pressure changes to one layer. Samples are prepared using leveling jigs to achieve a uniform layer thickness (5 mm), weighted to measure initial density, and jacketed to separate the sample and pore fluid from the confining oil (Fig. 7c); details in ref. [58].

Confining cell pressure ($P_c$) and up-/down-stream pore pressures ($P_{pu}$ and $P_{pd}$) are applied by three independent hydraulic servo-controlled intensifiers (Fig. 7a) and measured via diaphragm pressure transducers (accuracy: ±7 kPa) and LVDT displacements transducers. Confining pressure is applied using hydrogenated paraffinic white oil (vaseline oil viscosity ISO15), while pore pressure is applied using water. For our sample geometry the effective normal stress ($\sigma'_n$) on the gouge layers is:

$$\sigma'_n = (\sigma_n + P_c) - P_p \qquad (1)$$

where $\sigma_n$ (MPa) is the normal stress applied from the horizontal ram, $P_c$ (MPa) is the confining pressure and $P_p$ (MPa) is the equilibrated pore pressure in the sample.

All output signals are recorded using a simultaneous multichannel A/D converter with 24 bit/channel resolution at a sampling rate of 10 kHz and averaged for storage at rates between 1 Hz and 10 kHz. All the experiments were recorded at a minimum recording rate of 10 Hz up to 1000 Hz.

We performed experiments on the fraction <600 μm of sample B4 to study: (1) shear zone strength and permeability at in situ stress conditions; (2) rate-and-state frictional properties for shear displacement rates typical of real rockslide slow-to-

fast transition; (3) the influence of pore-pressure changes on the time-dependent slip behavior of the shear zone.

To determine these properties, we used different experimental protocols:

Stable-sliding shear experiments: we measured the stable-sliding frictional strength and permeability of the saturated shear zone material at constant $\sigma'_n$ of 2, 3, and 4 MPa. Tests were conducted at constant shear displacement rate of 10 μm s$^{-1}$, and displacements >10 mm were reached to localize shear deformation and achieve residual strength. Experiments allowed obtaining the residual Mohr-Coulomb failure envelope of the sample material at in situ stress conditions (Fig. 2a). After the sample attained a constant shear strength and the shear fabric was formed, we measured sample layer permeability in quasi-static loading conditions (i.e., the vertical ram was stopped) using the constant head method. We imposed a pressure gradient (1 MPa) between the up- and down-stream pore fluid intensifiers and measured the resulting flow rate across the sample. We calculated the sample saturated hydraulic conductivity ($K$, m s$^{-1}$) and intrinsic permeability ($k$, m$^2$) using the Darcy's law and assuming a water viscosity of $8.9 \times 10^{-4}$ Pa s (room temperature).

Velocity-step experiments: we evaluated the rate-and-state frictional properties of the sample material by velocity-step tests[59] performed outside the pressure vessel at room temperature and relative humidity (~30%) in dry and saturated conditions. Saturated conditions were achieved by surrounding the sample with an impermeable flexible membrane filled with water and left for ~1 h under a normal load of 0.5 MPa. Experiments were carried out with constant $\sigma'_n$ of 2 MPa. Each experiment started with a first stage where the sample was sheared at constant displacement rate of 10 μm/s until residual shear strength was achieved (Fig. 8a). Then a computer-controlled velocity-step history was imposed, with sliding velocity ranging from 1 to 300 μm/s and a total displacement for each velocity step of 500 μm. Rate-and-state friction parameters were obtained by modeling each

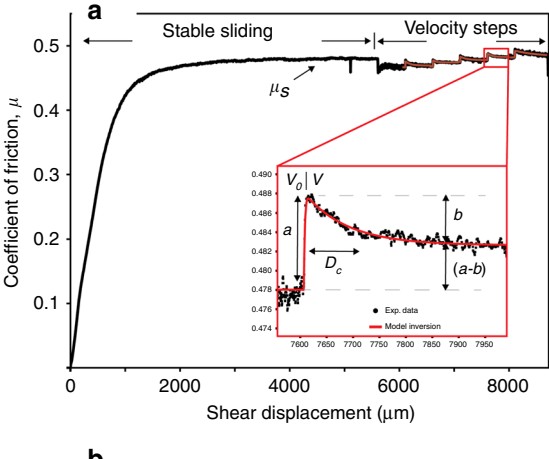

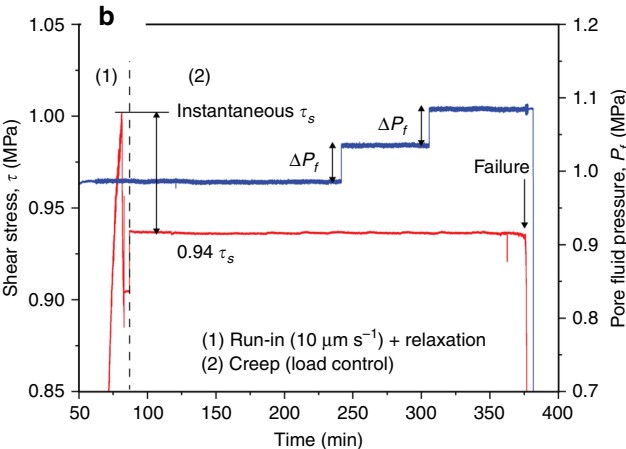

**Fig. 8 Experimental procedures. a** Displacement control: stable-sliding and velocity steps; **b** load control: creep experiments.

velocity step according to the general rate-and-state friction constitutive and state evolution equations[60,61]:

$$\mu = \mu_0 + a\ln\left(\frac{v}{v_0}\right) + b\ln\left(\frac{\theta v_0}{d_c}\right) \qquad (2)$$

$$\frac{d\theta}{dt} = -\frac{v\theta}{D_c}\ln\left(\frac{v\theta}{D_c}\right) \qquad (3)$$

where $\mu_0$ (dimensionless) is a reference coefficient of friction at sliding velocity $v_0$ ($\mu$m/s), $v$ is the frictional slip rate, $a$ and $b$ are empirical dimensionless constants, $D_c$ ($\mu$m) is the critical slip distance (i.e., distance required to renew asperity contacts), and $\theta$ (s) is the state variable (i.e., the average contact lifetime). Rate-and-state parameters $a$ and $b$ (Fig. 2b) were measured for dry and saturated conditions by solving Eqs. (2) and (3) with a 5th order Runge-Kutta numerical integration scheme with adaptive step size. Best-fit values are determined using an iterative least-square method. For a typical model fit, the unweighted chi-square error is usually ≤0.0001 and the variance is ≤5 × 10$^{-7}$. Estimated error is expressed as one standard deviation, usually ≤0.0002 (smaller than experimental uncertainties)[62,63].

Creep experiments: we interrogate the time-dependent response of the rockslide basal shear zone to short-term pore-pressure changes (i.e., groundwater recharge due to intense rainfall or rapid snowmelt events). We deformed gouge samples under in situ stress conditions in non-conventional pore-pressure-step creep experiments, in which the pore pressure is modulated following different stress paths stepwise (Fig. 2c). Creep experiments were carried out within the pressure vessel after the initial application of a constant confining pressure ($P_c$) of 2 MPa and a pore pressure ($P_p$) of 1 MPa. We started by applying the confining pressure of 1 MPa and allowed for sample compaction. The applied normal stress was then increased to the target value of 1 MPa and maintained constant throughout the experiment. At this stage, the up-stream pore-pressure intensifier was advanced to apply a small pore pressure, generally 0.2 MPa, while the down-stream intensifier was left open to the atmosphere until flow through the gouge layer was established. Once we ensured that gouge layers were fully saturated and all the residual air in the gouge was expelled, the

down-stream intensifier was closed to the atmosphere and left to equilibrate with the $P_{pu}$. Pore pressure was then increased to the target value. The sample was left to equilibrate for about 30 min while compaction occurred, and the layer reached a steady-state thickness. Each experiment begun after that we ensured fully saturation of the sample and under fully equilibrated pore-pressure conditions (Fig. 8b). Each experiment consisted in an initial stage (Fig. 8b, stage 1) where the sample was sheared at constant displacement rate (10 $\mu$m/s) to achieve shear localization and instantaneous strength (run-in stage). Then by stopping shear the specimen relaxed and reached a residual shear strength (hold stage). At this point, the actual creep stage was started (Fig. 8b, stage 2) by switching the vertical ram to load-control mode to maintain a constant shear stress and monitor the resulting evolution of slip. In each experiment, shear stress was set to a constant value corresponding to a mobilized shear strength of 86%, 89%, and 95% of the instantaneous strength, respectively (Fig. 2c). Then, while keeping shear stress constant, pore pressure $P_p$ was increased stepwise of about 0.05 MPa (corresponding to a rockslide groundwater table rise of about 5 m) and kept constant for at least 1 h until the onset of secondary creep or frictional instability (i.e., tertiary creep and collapse). In each experiment, we monitored normal stress, shear stresses, and displacement (either in the horizontal direction to measure sample dilation/compaction and vertical direction to measure slip), as well as fluid pressures ($P_c$, $P_{pu}$, $P_{pd}$) with time (Fig. 3, Supplementary Figs. 3 and 4).

**Analysis of laboratory and in situ hydro-mechanical response.** Both experimental and in situ monitoring data were analyzed using an ad hoc procedure to: (a) unravel the hydro-mechanical response of the rockslide basal shear zone material to short-term pore-pressure changes; (b) compare laboratory and in situ data quantitatively.

For each pore-pressure increment step of each creep experiment carried out in the pressure vessel (Figs. 2c, 3, Supplementary Figs. 3 and 4), we characterized the magnitude (i.e., shear displacement rate) and timing of the undrained hydro-mechanical response of the system using the following descriptors (Fig. 5a): (a) peak velocity ($V_P$): the maximum value of shear displacement rate attained in a short-term acceleration pulse following each stepwise increase of $P_p$; (b) long-term velocity ($V_{CII}$): the steady-state velocity attained during the Creep II, following each stepwise increase of $P_p$, accompanied by compaction; (c) time to peak velocity ($TV_P$): time between the stepwise pore-pressure increase and the velocity peak; (d) time to steady-state velocity ($TV_{CII}$): time between the stepwise pore-pressure increase and the attainment of Creep II. All times were normalized to $TV_{CII}$.

The same descriptors were evaluated for selected acceleration periods of the Spriana rockslide (Fig. 1c; Supplementary Table 1) for comparison to laboratory data. To this aim we considered time series of rockslide displacement and groundwater level, consisting of 2808 daily measurements acquired between 24/05/1996 and 28/09/2004[45,64] (courtesy ARPA Lombardia) and encompassing periods of different magnitude and timing of rockslide activity. Displacements were measured by a surface wire extensometer (E106) located near the rockslide upper scarp, while groundwater level was recorded by a hydraulic piezometer (PZ106) installed in a borehole reaching the basal shear zone near the rockslide toe (Fig. 1c). Rockslide instantaneous displacements rates were calculated as the three-point 1st derivative of the cumulative displacement time series. Groundwater levels were converted into pore-pressure values at the rockslide shear zones under the hydrostatic assumption: $P_p = \gamma \cdot h$ ($\gamma$ = unit weight of water = 9.8 kN m$^{-3}$; $h$ = measured water level above the shear zone). Eight periods of rockslide activity, responding groundwater table rises between 0.6 and 3.24 m (i.e., pore-pressure increment in the shear zone between 0.006 and 0.032 MPa) were considered (Fig. 1c). Evaluating the timing of rockslide response was biased by undefined delays related to the different positions of surface displacement and groundwater measurement devices, resulting in noisy data. Thus, the comparison between in situ and laboratory data (Fig. 5b–d) was mainly based on displacement rates.

We applied non-parametric statistics to attempt a quantification of the occurrence and strength of data trends in terms of the Spearman's rank correlation coefficient, $\rho$[65] (Supplementary Table 2). Non-parametric statistics allows to assess the occurrence of correlation between data independently on data normality and linearity of the regression model, and provide a conservative evaluation of correlations[66]. After checking that all the variables satisfy the normality assumption (Kolmogorov–Smirnov test), when we found very high values of the Spearman's coefficient (1 or close to 1), we also quantified the parametric Pearson's correlation coefficient, $r$ (Supplementary Table 2). We did not quantify correlation coefficients for laboratory data from creep experiments with $\tau_{mob} = 0.95$ (too few data).

## Data availability

The raw experimental datasets generated and analyzed during this study are available as supplementary Source Data. Other data are available in raw or table format from the corresponding author upon request.

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

## Acknowledgements

We thank ARPA Lombardia (Dr. Luca Dei Cas, Dr. Gregorio Mannucci) for logistic support at the sampling site and for providing monitoring datasets, Dr. Antonio Piersanti for support at the INGV HP-HT laboratory facility, and Dr. Valentina Barberini for XRPD analyses. This research was partially supported by Fondazione Cariplo Grant 2016-0757 "Slow2Fast" to Federico Agliardi; ERC grant Nr. 259256 "GLASS" to Cristiano Collettini; and Marie Sklodowska-Curie grant No. 656676 "FEAT" to Marco M. Scuderi.

## Author contributions

F.A. and C.C. conceived the research. F.A. and N.F. collected the samples and performed basic lab characterization. All the authors contributed to the experimental design. M.M.S. performed the experiments and processed raw experimental datasets. F.A. and M.M.S. analyzed experimental results and in situ monitoring data. F.A., C.C., and M.M.S. wrote the paper.

## Competing interests

The authors declare no competing interests.
