## [Peer Review File · Nature Communications]

Reviewers' comments:

Reviewer #1 (Remarks to the Author):

Dear Editor,

Please find my review of the manuscript entitled:

Slow to fast transition of giant creeping rockslides modulated by undrained loading in basal shear zones

By Agliardi et al.

The paper is related to the understanding of creep processes the path to catastrophic failure of rockslides. A comparison between experiments on sample coming from the Spriana landslide and its behaviour is proposed. The observation of the 3 traditional creep stages is realized increasing the fluid pressure increment, which induce acceleration and reaching a steady steady after deceleration. It shows the importance of groundwater effect (in undrained condition) with time and the dependence on the long-term velocity, that can lead to catastrophic failure. It provides an explanation for giant rockslide failure.

General comments

This paper is very interesting it provides important results about rockslides catastrophic failure. It shows how to reach catastrophic failure in gauges. It is very interesting that they were able to reproduce such fast acceleration and deceleration which is very common for landslides. That is why it is an important topic. Such experiment lies in between landslide sciences and seismology. It is also very stimulating because it opens a lot of questions:

- How many steps are necessary to reach failures, is it variable?
- The 3 stages are they always observed.
- It seems that most important figure is the 4b, can this velocity be reach in one step?

You must know that I am not a fan of the 3 stages creep. It those not mean that I am opposed to the results of this paper. In my opinion, the results with figure 4b are provide more general results. It would be great if the author can underline this point. In figure 4b, it would be nice to put different shapes for the points for the different Tau mob, but put the colour scale for the pressure, or in the background. The colours are also difficult to follow in the figure 4 d, e, f, maybe follow this proposal.

I am confident that this paper is a valuable contribution for Nature communications.

Specific comments:

Line 37: the driving force does not only glaciation and deglaciation, you can add river incision that create steep slopes, in a uplifting context.

Figur1: in b the blue filled area is the ground water? If yes add a legend and related date of the situation. Indicate d in a better way on c.

Lines 96-97: explain how to consider that a wire extensometer is representative of the failure surface deformation...

Lines 111 -115: for the community of landslide please indicate the corresponding friction angle.

Line 241: 0.89 corresponds to τ_s ?

Line 321: σ_n , n in subscript (take care along the text with this problem Pp and pc lines 367-368).

Figure 6: can be moved to supplementary material?

For equation 2 and 3: provide units.

Fig S1a: some indication about the sampling distance to basal surface of a small scheme must be created to locate the relative positions.

Fig S2: equal scale for the stress, which allows to get the friction angle.

Figures S3 and S4: it would be great to have the same colour scale as figure 2 and 3, it is a bit confusing.

Reviewer #2 (Remarks to the Author):

Review of Agliardi et al., "Slow to fast transition of giant creeping rockslides..."

This manuscript describes laboratory tests on landslide slip surface materials and in-situ monitoring to infer the processes that often lead to acceleration and catastrophic failure of large bedrock landslides. The lab tests clearly show that the landslide material initially dilates when it begins to shear, causing temporary strengthening via a reduction in the effective stress, but then compacts and weakens with continued shear. If the pore pressure that triggers failure is raised high enough, this can lead to runaway acceleration. That dilation and compaction of low permeability slip surface materials can regulate landslide motion has been well established theoretically, but lab tests illustrating this behavior are still rather rare. The manuscript then describes in-situ observations of pore pressure and displacement and shows that several key behaviors are consistent with lab tests, specifically the magnitude and timing of velocity increases in response to elevated pore pressures.

This is a novel study, as it combines detailed lab measurements on actual landslide slip surface material with detailed in-situ observations of that landslide's behavior. Many studies have done one or the other, but connecting behavior between the lab and field scales continues to be challenging. Although the comparison between lab and field measurements is not quite direct (the authors use general characteristics of the velocity response to changes in pore pressure rather than directly forward modeling the landslide's motion with the lab-determined frictional parameters), I think the overall correspondence between lab and field behaviors is fairly convincing. My main critique of the manuscript that I would like to see addressed in a revision is that that correspondence should be clearly demonstrated objectively with statistics, rather than qualitatively interpreted as done in the current manuscript. Basically, any statement that claims that a certain behavior is the same or different in the lab vs. in-situ data should be backed up with a relevant statistic, as should any statement about whether there is or is not a trend in the data. This comment mostly applies to Fig. 4 and associated text. I anticipate the authors should be able to address this concern in a revision, and therefore recommend minor revisions. A few additional comments I would like to see addressed are given by line number below:

36: Please define "sub-critically stressed." Factor of safety approximately 1?

55: I would tone down the language here. I think the basic physical mechanisms of dilation/compaction and resulting changes to pore pressure are well understood, but it's the lab and field measurements of those physical mechanisms that are lacking.

68: Please insert a brief description of rate-and-state friction here. The results include lab tests to determine parameters for the rate-and-state model, so it should be included in the Introduction. Also, I would recommend adding a sentence or two acknowledging other proposed mechanisms that may explain runaway landslide acceleration, e.g. growth of a shear fracture to a critical nucleation size (Viesca and Rice, 2012, JGR-solid earth), or micro-cracking of ductile materials (Petley and Allison, 1997, ESPL).

169-71: A qualitatively similar velocity pattern was observed over the course of a few months leading up to catastrophic failure of the Mud Creek landslide in California as observed by

Handwerger et al., 2019 (Reference #24, see their Fig. 3). Is that velocity pattern leading up to rapid acceleration unique to the dilatancy followed by compaction mechanism, and would it be reasonable for that mechanism to operate over monthly time scales? I ask because if it is a unique pattern, there may be important implications for forecasting catastrophic landslide acceleration from velocity time series data. This is quite challenging to do with the traditional 1/velocity technique, as it produces a lot of false positives. I would recommended coming back to the issue of landslide forecasting in the Discussion section later.

195: Fig. 4d-f: Please use statistics to objectively state whether field data do or do not have a significant trend (does the regression slope significantly differ from 0?), and/or whether or not the field data are different or not compared to the lab data (t-test should be fine for this). I don't suspect it will change the main findings of the study much, but I don't think it's sufficient to just plot the data and qualitatively interpret whether or not they are consistent.

229-38, and 263: Support these statements with relevant stats and modify if needed.

375: Earlier in the paper the rates are given as 1-500 $\mu\text{m/s}$.

Reviewer #3 (Remarks to the Author):

The article reports laboratory measurements of the frictional strength of rock material coming from an active rockslide and uses the experimental results to compare with natural data of slip and groundwater level in the active rockslide of Spriana (Italy). The new information comes from linking the laboratory data to the natural data and progress on the prediction of landslide failure.

In the experiments both the slip rate and the pore fluid pressure are varied and results show that 1) the material is velocity strengthening, 2) increasing pore fluid increases the slip date and may drive the interface toward catastrophic failure. To my knowledge the combination of these two effects and their role in landslide stability has not been demonstrated in previous studies and is innovative.

However, what is lacking, and hopefully the authors could come up with some proposition, is a strong conclusion of the study that would deserve publication in a high impact journal. Lots new of field and experimental results are presented, but how these data change our view on landslides and how results may change predictions on the rockslide in Spriana are buried in many other information. This would make this article more impactful to put up-front one major conclusion. Several points should be answered before this study could be published in a high impact journal such at Nature Communications.

1) Some literature has been overlooked. Handberger et al. PNAS, 2016 (vol. 113, 10281-10286), used rate-and-state friction to explore landslide stability and shown that adding elastic forces in the system control as well the stability. This effect of elasticity should be discussed by the authors of the present study (and this could be the main difference between the experimental results where not so much elastic energy is stored in the system, compared to a natural landslide). Helmstetter et al. (Journal of Geophysical Research, 2004 (vol. 109), B02409) have shown theoretically that creep acceleration of landslides could occur both in the velocity-weakening and velocity-strengthening regimes.

2) The static friction coefficient is measured equal to 0.51 (line 111 and following). Some of the conclusions of the experimental data, applying the rate-and-state friction law implicitly assumes that the static friction coefficient is constant over the range of velocities 1-300 $\mu\text{m/s}$ (whereas the dynamic friction coefficient will vary). If the strength of the landslide material in the experiment varies significantly, the interpretation of the author on the origin of the strengthening

effect would be different. One way to visualise such effect would be to report the steady-state strength of the interface in the experiments for the range of velocities 1-300 micron/s and add this graph in supplementary material. Said differently, the authors have to ensure that the Mohr-Coulomb envelope of Fig. 2a is independent on the loading rate of the interface, in the range of loading velocities used in the experiments.

For example, Thøgersen et al. (Geophysical Research Letters, 2019, doi: 10.1029/2019GL084436) have recently demonstrated that slow slip rates in faults are transient and controlled by the pre-stress on the interface and viscous/frictional dissipation (i.e. how the strength evolves with sliding velocity). This article has been published probably after the authors have submitted their work to Nature Communications, but the discussion there could help the authors explain their results.

3) An important outcome of the study, that would argue for publication in a high impact journal, would be to show predictions on the behavior of the Spriana rocks slide. For this, field data should be plotted in Fig. 4b and a discussion on what range of parameters or pore fluid pressure increase would induce a catastrophic failure of this landslide would be interesting. I suggest that the authors use their (high quality) data to discuss potential predictions or scenarios on how the rockslide could evolve in the future.

Minor points

Figure 1C: It is not clear how the different rockslide acceleration periods are defined. For example events 6 and 7 do not correspond to a significant increase of the deformation rate (the water-blue data are not correlated to the deformation-gray data). And what is the criterion to separate event 6 from event 7 ? Looking at the data, there is a high water level period between events 4 and 5 and no clear acceleration of the landslide deformation. Any idea why ?

In addition, I may suggest reporting the correlation coefficient between the water level and ground deformation and define events due to fluid increase pressure when the correlation coefficient is above a certain threshold.

Line 69: remove «innovative» (several experimental results using this machine have been already published. In the present study, the material is new, not the experimental set-up).

Line 76: enucleated -> nucleated

Line 191: The sentence "slip rates larger than the hydraulic conductivity" does not make sense. Even if slip rate and hydraulic conductivity have the same dimensions (m/s), they represent two different processes and their values cannot be compared to each others.

Line 241: "shear strength of 0.89", what is the unit (MPa) or is it normalized (like a friction coefficient)?

Reply to Comments from Reviewer #1

General Comment 1: How many steps are necessary to reach failures, is it variable? The 3 stages are they always observed. It seems that most important figure is the 4b, can this velocity be reach in one step?

Reply: We thank the reviewer for raising these points. Any form of brittle creep, in rock or in granular material, occurs in subcritical conditions, i.e. when a fraction of the instantaneous / ultimate strength of a material is mobilized. In the studied gouge material, the occurrence of creep, the changes in steady-state creep rates and the onset of non-linear (tertiary) creep depend on the interplay between processes favouring instability (i.e. applied shear stress, pore pressure control on effective normal stress in undrained and “semi-drained” conditions), strengthening processes related to the intrinsic rheological response of the material to slip acceleration (i.e., velocity strengthening behavior) and dilation strengthening of the pressurized material. As we state at Lines 221-225 (revised manuscript), we infer that the three creep stages observed in the lab and *in situ* are the macroscopic manifestation of the competing processes mentioned above, with steady-state creep (Creep II) representing a condition of dynamic equilibrium rather than a specific process itself. On the other hand, the occurrence of decelerated creep (Creep I) in granular material appears related to the hydro-mechanical effects of fluids (undrained dilatant strengthening).

The number of fluid pressure incremental steps thus depend on: a) the mobilized stress related to slope topography, strength and stiffness (“how close we are to the critical conditions for instantaneous failure”); b) mobilized stress change (or rate of change) due to slope loading, toe erosion or slope shape modifications; c) the magnitude and rate of pore pressure perturbations related to groundwater rise, e.g. induced by intense rainfall or rapid snowmelt. Therefore yes, of course failure can be reached in one step if shear stress and pore pressure perturbations result in critical effective stress conditions. We have mentioned this in the revised version of the manuscript (now Lines 226-230).

General Comment 2: In my opinion, the results with figure 4b are provide more general results. It would be great if the author can underline this point. In figure 4b, it would be nice to put different shapes for the points for the different Tau mob, but put the color scale for the pressure, or in the background. The colours are also difficult to follow in the figure 4 d, e, f, maybe follow this proposal

Reply: We thank the reviewer for the suggestion. We tried to adapt the figure according to the suggestions, but we found that the figure is more effective in the present version because: a) in Figs 4 b,d,e,f we use dots for laboratory data and stars for in situ data (now added also in Fig. 4b), thus we prefer to keep colours to distinguish between different levels of Tau mob; b) pore pressure values for different pressure steps do not coincide exactly in the different experiments, and pore pressure values are already explicit in the plot abscissa. Anyway, we moved the legend in a more visible position (to inset f) and added some explanatory labels.

Comment to Line 37: the driving force does not only glaciation and deglaciation, you can add river incision that create steep slopes, in a uplifting context.

Reply: We agree, and we have added some text.

Comment to Figure 1: in b the blue filled area is the ground water? If yes add a legend and related date of the situation. Indicate d in a better way on c.

Reply: Yes, the blue filled area is a representation of the perched aquifer based on the integration of borehole and piezometers observations made upon 1989-90 site investigations. In the revised version we have better clarified this by labelling in bold the perched aquifer and mentioning it in the figure caption.

Comment to Lines 96-97: explain how to consider that a wire extensometer is representative of the failure surface deformation.

Reply: This is a good point. At the end of the Introduction, we stated that “The rockslide mass is affected by relatively small internal deformation, so that long-term surface displacements are mainly related to hydro-mechanical forcing along the basal shear zone”. This assumption clearly carries simplifications but is supported by the analysis of subsurface investigation data (high quality drill cores; borehole inclinometer measurements performed after 1989 site investigations; topographic, extensometer and distometer baseline surface displacements data). These data have already been published and discussed elsewhere (Refs 9, 42, 43) and have not been presented here. We have now better specified this point in the revised version of the manuscript.

Comment to Lines 111-115: for the community of landslide please indicate the corresponding friction angle

Reply: Done.

Comment to Line 241: 0.89 corresponds to τ_s ?

Reply: There was a typo: corrected “shear strength of 0.89” to “shear strength of $0.89 \tau_s$ ”

Comment to Line 321: σ_n , n in subscript (take care along the text with this problem Pp and pc lines 367-368).

Reply: Done and thank you for the suggestion.

Comment to Figure 6: can be moved to supplementary material?

Reply: We have thought about the most suitable position for this figure and finally we have decided that having this technical figure in the Methods section (thus separated from the main text but readily available to the reader) can improve the clarity of the experimental data. This also because the adopted experimental approach in load control during fluid pressure stimulation of experimental sliding surfaces is a quite new method also within the rock deformation community.

Comment to Equations 2 and 3: provide units

Reply: Thank you for the comment. Done.

Comment to Figure S1a: some indication about the sampling distance to basal surface of a small scheme must be created to locate the relative positions.

Reply: the position and depth of the studied samples is portrayed in Fig. 1b and described in detail in the first paragraph of the Methods section. About “sampling distance to basal surface”: actually, all the samples were collected within the basal shear zone at different locations along the slope as clearly shown in Fig. 1b. We have now better specified this point in the figure caption: “(a) Grain size distributions of cataclastic shear zone samples collected from the basal shear zone of the Spriana rockslide (locations in Fig. 1b).”

Comment to Figure S2: Fig S2: equal scale for the stress, which allows to get the friction angle.

Reply: Thank you for the comment. We have revised Fig. S2 accordingly.

Comment to Figure S3 e S4: it would be great to have the same colour scale as figure 2 and 3, it is a bit confusing.

Reply: Thank you for the comment. We have edited Figures S3 and S4 accordingly.

Reply to Comments from Reviewer #2

General Comment: This is a novel study, as it combines detailed lab measurements on actual landslide slip surface material with detailed in-situ observations of that landslide's behavior. Many studies have done one or the other, but connecting behavior between the lab and field scales continues to be challenging.

Reply: Thanks for having highlighted the innovative and challenging aspect of our work.

General Comment: My main critique of the manuscript that I would like to see addressed in a revision is that correspondence should be clearly demonstrated objectively with statistics, rather than qualitatively interpreted as done in the current manuscript. Basically, any statement that claims that a certain behavior is the same or different in the lab vs. in-situ data should be backed up with a relevant statistic, as should any statement about whether there is or is not a trend in the data. This comment mostly applies to Fig. 4 and associated text. I anticipate the authors should be able to address this concern in a revision, and therefore recommend minor revisions.

Reply: We agree and thank the reviewer for his comment. In general, statistics has been applied in all the stages of the experimental investigation, both to process and evaluate our data (Methods and Figures): a) grain size and mineral composition of different fractions of the gouge samples (CDF for grain size and related percentiles; use of box plots to compare the variability / sensitivity of mineral composition; Fig. S1); b) characterization of the friction coefficient of the tested gouge material and its dependence on grain size (statistical fit and confidence limits to evaluate the sensitivity of the friction coefficient to the tested grain size fraction; Figs. 2 and S2); c) processing (filtering / smoothing) of raw experimental data and rate-and-state modelling (see Methods and Figs. 2 and 7); d) analysis (filtering / smoothing) of *in situ* displacement time series to extract the different displacement rate and timing parameters (V_p , V_{cII} , TV_p ; TV_{cII}) used to compare. This allowed us to obtain clean and clear experimental data, resulting in a high-resolution analysis of slip displacements, displacement rates, dilatancy and their relative timing (Figs. 3 and 4a – now 5b).

The results in Figs. 4b/d/e/f (now 5b, c d, e), on the other hand, are based on the synthetic parameters mentioned above (TV_p , V_p , V_{cII} vs. ΔP_f). They are evaluated for each pore pressure step increment of the experiments carried out at different values of mobilized shear strength. These data, while identifying clear lab and *in situ* trends, consist of only few points (see new Table S2 below), implying that: a) regression models are unknown; b) there is a poor chance to describe any trend in a statistical significant way.

Table S2 – Correlation statistics for laboratory and rockslide data trends in Fig. 5.

Variables	Dataset	# data	Spearman ρ	p-value	Perason r	p-value
$T_{VP} - \Delta P_f$ (Fig. 5b)	Lab: $\tau_{mob} = 0.86$	3	-0.5	0.67	-	-
	Lab: $\tau_{mob} = 0.89$	4	-1	-	-0.93	0.07
	Lab: $\tau_{mob} = 0.95$	2	-	-	-	-
	Rockslide	8	-0.53	0.17	-	-
$V_P - \Delta P_f$ (Fig. 5c)	Lab: $\tau_{mob} = 0.86$	3	1	-	0.997	0.046
	Lab: $\tau_{mob} = 0.89$	4	0.8	0.2	-	-
	Lab: $\tau_{mob} = 0.95$	2	-	-	-	-
	Rockslide	8	0.13	0.75	-	-
$V_{CH} - \Delta P_f$ (Fig. 5d)	Lab: $\tau_{mob} = 0.86$	3	1	-	0.928	0.24
	Lab: $\tau_{mob} = 0.89$	4	1	-	0.912	0.08
	Lab: $\tau_{mob} = 0.95$	2	-	-	-	-
	Rockslide	8	0.92	0.001	0.9	0.002

That is the reason why we first stuck on a qualitative evaluation of the experimental vs *in situ* trends. Nevertheless, we find the reviewer’s suggestion appropriate and stimulating. We try now to quantify the robustness of observed trends better. Although data are limited for allowing a robust definition of specific regression functions, we applied non-parametric statistics to quantify the occurrence and strength of data trends in terms of the Spearman’s rank correlation coefficient, ρ (Spearman, 1904; Table S2). Non-parametric statistics allows to assess the occurrence of correlation between data independently on data normality and linearity of the regression model and provide a conservative evaluation of correlations (Swan and Sandilands, 1995). After checking that all the variables satisfy the normality assumption (Kolmogorov-Smirnov test), when we found very high values of the Spearman’s coefficient (1 or close to 1), we also quantified the parametric Pearson’s correlation coefficient, r (new Supplementary Table S2 above). We did not quantify correlation coefficients for laboratory data from creep experiments with $\tau_{mob} = 0.95$ because of very few data.

The results confirm our previous observations and allow to draw some useful insights:

- short-term systems response: laboratory values of time to peak velocity (T_V) measured in the laboratory shows a negative correlation with pore pressure stepwise increase (ΔP_f), but the correlation is weaker at low values of τ_{mob} . Rockslide *in-situ* T_V data are noisy and weakly correlated with (ΔP_f) ($\rho=-0.53$). Similarly, peak velocity (V_P) measured in the lab shows a strong positive correlation with ΔP_f ($\rho=0.8-1$), while rockslide data are noisy and characterized by a weaker positive correlation. In any case, rockslide values of T_V and V_P quantitatively fall within the lab experimental bounds and show the same general trends. The weakness of *in situ* data trends suggests that the short-term response of real rockslides to fast-pore pressure changes is noisy, due to a number of active physical processes and controlling factors interacting in a short time. That is why it is very difficult to establish simple empirical relationships between groundwater levels and rockslide displacements/rates and the short-term response should be considered as a weak predictor of rockslide evolution to collapse.

- long-term systems response: both laboratory and *in situ* values of steady-rate creep velocity (V_{CII}) show a very strong positive correlation ($\rho=1$, in some cases close to statistical significance) in a quite robust log-linear form (Pearson's r exceeding 0.9). Moreover, rockslide trend shows a statistically significant similarity with the trend of experimental data at $\tau_{mob.} = 0.89$ (t-test t-value: -1.483; $Prob > |t| = 0.19$). These strong correlation trends suggest that long-term velocity V_{CII} and ΔP_f can be robust descriptors to establish empirical threshold conditions for transition to collapse, and support the plot in Fig. 5e (Figure 4b in the previous version of the manuscript).

We edited the Methods and the Discussion sections (lab vs. *in situ* comparison; implications for prediction) to account for the analyses and considerations above, according to the suggestions proposed by the Reviewer's and the Associate Editor.

Comment to Line 36: please define "sub-critically stressed." Factor of safety approximately 1?

Reply: as mentioned in the response to Reviewer 1 above, in subcritical stress conditions a fraction of the instantaneous / ultimate strength of a material is mobilized (see for example Riva et al., 2018 – ref 9). The term "subcritical" is borrowed from fracture mechanics (e.g. Atkinson and Meredith, 1987), and also commonly used in the landslide community when dealing with progressive rock slope failure (e.g. Eberhardt et al., 2004; Stead et al., 2006; Lacroix and Amitrano, 2013; Riva et al., 2018). In continuum mechanics, subcritical conditions can be expressed in terms of fractions of instantaneous shear strength, uniaxial compressive strength, or differential stress (triaxial conditions). In any case, they correspond to a "factor of safety" slightly larger than one, although the factor of safety is a limit equilibrium static concept that only apply to rigid-plastic o elasto-plastic material behaviours.

Comment to Line 55: I would tone down the language here. I think the basic physical mechanisms of dilation/compaction and resulting changes to pore pressure are well understood, but it's the lab and field measurements of those physical mechanisms that are lacking.

Reply: We thank the reviewer for this comment. Actually, we think that the two aspects are related. As we reported in the introduction, sudden or delayed acceleration pulses, long periods of sustained steady displacement rates, or runaway rupture resulting in catastrophic collapse have been commonly observed in creeping landslides. Nevertheless, it is very difficult to find reliable statistical correlation between the magnitudes, rates and timing of the landslide triggers (e.g. rainfall, snowmelt) and the landslide response (displacements, displacements rates). Moreover, analytical and numerical models used to analyse landslide mechanisms are usually unable to reproduce the full spectrum of observed time-dependent behaviours. This is due to the fact that, although the underlying physical mechanisms are well understood (as suggested by the reviewers), the mechanisms acting *in situ* within large landslides and their interplay are not so clear. We have now tried to specify it better in the text.

Comment to Line 68: Please insert a brief description of rate-and-state friction here. The results include lab tests to determine parameters for the rate-and-state model, so it should be included in the Introduction. Also, I would recommend adding a sentence or two acknowledging other proposed mechanisms that may explain runaway landslide acceleration, e.g. growth of a shear fracture to a critical nucleation size (Viesca and Rice, 2012, JGR-solid earth), or micro-cracking of ductile materials (Petley and Allison, 1997, ESPL).

Reply: We thank the reviewer for the comment. We edited our introduction to explain better that:

- a) creep in large rock slopes is the macroscopic evidence of different processes. In the early stage of development of rock slope instability, creep reflects crack damage accumulation in subcritical stress

conditions (i.e. “progressive failure”), leading to localization and gradual formation of a localized basal shear zones. This is well illustrated by the cited papers based on experimental evidence and mathematical modelling. The cited manuscripts are more recent and focused on large rock slope stability than Petley and Allison, 1997, thus we avoided including this reference (also due to a limit in the reference number). Then, once the basal shear zone has accumulated enough shear strain to reach maturity (i.e. thickness, lateral continuity, reduced grain size, low permeability) due to cataclastic processes, creep becomes dominated by the hydro-mechanical behaviour of the shear zone and its sensitivity to sharp changes of hydrological conditions.

- b) the main problem with large creeping rockslides is to establish physically-based criteria to predict the change of behaviour from steady, slow creep to the fast transition to catastrophic failure. Empirical and statistical methods usually fail to capture the full complexity and spectrum of rockslide creep styles. These can be explained by different mathematical models proposed in the literature, that indeed lack experimental and in situ validation. As suggested by the reviewer, we mentioned that rate and state models have been applied to large landslides (Handwerger et al., 2016, 2019), but did not include Viesca and Rice (2012) that is more focused on shallow submarine landslides in sediments. To better support the innovative character of our work, we also better specified that laboratory studies specifically supporting these advanced models in landslide science are lacking (they are usually based on material parameters derived by the fault mechanics literature).

Comment to Line 169-171: A qualitatively similar velocity pattern was observed over the course of a few months leading up to catastrophic failure of the Mud Creek landslide in California as observed by Handwerger et al., 2019 (Reference #24, see their Fig. 3). Is that velocity pattern leading up to rapid acceleration unique to the dilatancy followed by compaction mechanism, and would it be reasonable for that mechanism to operate over monthly time scales? I ask because if it is a unique pattern, there may be important implications for forecasting catastrophic landslide acceleration from velocity time series data. This is quite challenging to do with the traditional $1/\text{velocity}$ technique, as it produces a lot of false positives. I would recommend coming back to the issue of landslide forecasting in the Discussion section later.

Reply: Our *in situ* data (see new Supplementary Table S1 below), interpreted through their quantitative consistency with laboratory data, suggests that the impulsive undrained response to pore pressure increments and its modulation through dilatancy occur within few days to few weeks. On the other hand, periods of steady-rate creep (at velocities dependent on mobilized strength and pore pressure increment) last until pore pressure is maintained (Fig. 1).

Supplementary Table S1. Characteristics of selected rockslide acceleration periods (Fig. 1).

Acceleration period	Total Duration (days)	Time to peak velocity (TV_P, days)	Peak velocity (V_P, $\mu\text{m/s}$)	Long-term velocity (V_{CH}, $\mu\text{m/s}$)
1	32	3	0.067	0.003
2	35	3	0.0185	0.002
3	60	2	0.022	0.0017
4	47	5	0.031	0.0015
5	47	6	0.012	0.0025
6	18	5	0.012	0.0023
7	150	15	0.013	0.004
8	79	11	0.023	0.004

Regarding the mechanisms underlying the observed slip patterns, there are key differences between our work and that by Handwerger et al. (2019): a) the mineralogy of the landslide shear zone inferred by Handwerger et al. (2019) is not described by the authors, but considering the parent rock (Franciscan Complex) we can expect it to be significantly different from our gouge (quartzo-phyllsilicate mixture derived by gneiss); b) in Handwerger et al. (2019), pore pressures in the landslide are not measured or imposed, but modelled, thus there is no direct experimental control on their relationships with landslide response; c) in Handwerger et al. (2019), pore pressures rise and then decline after a peak, while in our experiments we maintain pore pressure constant and follow the spontaneous evolution of shear displacements and dilatancy/compaction. Then we use these observations to infer the underlying hydro-mechanical coupling processes.

Handwerger et al. (2019) say: “Catastrophic failure of the Mud Creek landslide is most likely due to frictional weakening that was initiated by large increases in pore-fluid pressure. These large stress changes resulted from extreme rainfall that followed a period of historic drought and may have been enhanced by increased pore-fluid pressures from shear-induced compaction and longitudinal compression of the upslope material due to differential slip. Our pore-fluid pressure model does not account for multi-dimensional pore-fluid pressure changes or for mechanical-hydrologic feedbacks such as shear-induced dilatancy or compaction, however this simple model has been widely used to describe the first-order changes in pore-fluid pressure that drive acceleration of similar deep-seated landslides around the world and indicates that the Mud Creek landslide experienced unusually high pore-fluid pressures prior to catastrophic failure.”

Thus, after listing a variety of possible mechanisms underlying the observed landslide behavior, they propose that periods of increasing velocity (eventually leading to catastrophic failure) could be related mainly to pore pressure increase (and effective stress reduction) mirrored by a rather direct relationship between pore pressures and displacement rates. In our experiments, we have observed a more complex mechanism of rockslide evolution, that is dominated by the interplay of short-term undrained effects (pore pressure increase \rightarrow effective stress reduction and sudden acceleration \rightarrow dilatant strengthening and deceleration) with long-term undrained to semi-drained effects (frictional sliding \rightarrow shear zone compaction and weakening until tertiary creep), that hamper an obvious direct correlation between groundwater levels and displacement rates.

We do not claim that our results are applicable to all the types of deep-seated landslides, that can be characterized by different materials, shear zone maturity, rheology, kinematics, and sensitivity to loading and fluid pressures due to different hydrological characteristics. However, we tested conditions (mineralogy, stress) representative of most large rockslides occurring in alpine setting dominated by crystalline basements. We added some text to the conclusions to better stress the implications of our results in a practical rockslide forecasting perspective.

Comment to Figure 4d-f: Please use statistics to objectively state whether field data do or do not have a significant trend (does the regression slope significantly differ from 0?), and/or whether or not the field data are different or not compared to the lab data (t-test should be fine for this). I don't suspect it will change the main findings of the study much, but I don't think it's sufficient to just plot the data and qualitatively interpret whether or not they are consistent.

Reply: See our extended reply to the General Comment of Reviewer #2 above. We have edited the text commenting Fig. 4 (now Fig. 5) accordingly.

Comment to Lines 229-38 and 263: support these statements with relevant stats and modify if needed.

Reply: See our extended reply to the General Comment of Reviewer #2 above. We have edited the related text accordingly.

Comment to Line 375: Earlier in the paper the rates are given as 1-500 $\mu\text{m}/\text{s}$.

Reply: Actually, the rates are given in the range 1-300 $\mu\text{m}/\text{sec}$ earlier in the paper and later in the Methods. Here, 1-500 $\mu\text{m}/\text{sec}$ was a typo, we deleted it.

Reply to Comments from Reviewer #3

We thank the Reviewer for the insightful comments that he/she provided.

General comment 1: Some literature has been overlooked. Handberger et al. (2016; PNAS vol. 113, 10281-10286), used rate-and-state friction to explore landslide stability and shown that adding elastic forces in the system control as well the stability. This effect of elasticity should be discussed by the authors of the present study (and this could be the main difference between the experimental results where not so much elastic energy is stored in the system, compared to a natural landslide). Helmstetter et al. (2004, Journal of Geophysical Research, vol. 109, B02409) have shown theoretically that creep acceleration of landslides could occur both in the velocity-weakening and velocity-strengthening regimes.

Reply: We thank the Reviewer for this comment since from one side it will help us improving the manuscript mentioning other studies dealing with landslides stability, and from the other side it will give us the opportunity to further highlight the innovative method we have adopted. In the following we firstly present the motivations that pushed us toward using a new method for laboratory experiments, then we will explain how and where we have improved the manuscript to account for this point, and finally we will provide some detailed comments on specific points presented in the papers Handwerger et al. (PNAS 2016) and Helmstetter et al. (2004).

1) The stability of sliding surfaces (faults or basal landslide shear zones) is commonly addressed in the framework of rate and state friction, R&S (e.g. Dietrich, 1979; Ruina, 1983; Marone et al., 1998 for a

comprehensive review). Rate weakening or strengthening represent second order variation in friction that are generally used to characterize the slip behaviour and indeed R&S friction has been proven successful in many case studies. However, when fluid overpressure develops within a sliding surface, the associated weakening produced by the reduction of the effective normal stress can be significantly larger than the rate-strengthening behaviour dictated by the frictional properties of the shear zone. Therefore, the slip behaviour of a sliding surface is strongly controlled by the coupling between fluid pressure development and the evolution of frictional properties with sliding. Due to technological limitations in: a) measuring R&S friction parameters at different level of fluid pressure and b) documenting fault slip evolution during pressurization, the interplay between fluid pressure development and evolution of frictional properties has not been properly characterized in laboratory experiments. In the present work we have filled this gap by developing new experiments to capture the hydromechanical coupling during fluid pressure builds up along the basal shear zone of a rockslide. We think that this is the best experimental approach to provide physically based foundations for the Spriana landslide since the landslide shows episodic or prolonged acceleration periods that are mainly controlled by fluid pressure development (Fig. 1c,d). This approach can be extended to other landslides where the slip behaviour is strongly connected with fluid pressure fluctuations along the basal shear zone.

2) In the revised version of the manuscript we have given credit to the previous works on landslide stability and we have emphasised the differences with our work at the end of the first paragraph of the discussion. In particular we have added the following paragraph: *“Our proposed mechanism can act in concert with other mechanisms proposed to explain the diversity in slip behaviour of landslides, i.e. accelerated creep vs. catastrophic failure, on the grounds of on rate- and state- frictional models (Helmstetter et al., 2004) and including also the force balance between the sliding surface and the surrounding loading medium (Handwerger et al., 2016). However, for landslides like Spriana where episodic or prolonged acceleration periods are mainly controlled by fluid pressure builds-up, our data indicate that the coupling between fluid pressure development and the evolution of frictional properties during sliding is the primary controlling factor in the slip behaviour”*.

3) The papers by Handwerger et al. (2016) and Helmstetter et al. (2004) present very interesting research topics on the same line of the work that we have been developing in the last 5 years. In this context, we would like to develop some thoughts to provide a better answer in this revision note, but we are not going to include in the main text since it is beyond the scope of the proposed research.

Handwerger et al. (2016) say: “we describe landslide motion using a rate- and state-dependent frictional model that incorporates a nonlocal stress balance to account for the elastic response to gradients in slip”. We totally agree with this and we have used a similar approach including, but not modelling like Handwerger et al., some experiments where by controlling the stiffness ratio between the fault and the surrounding (to control the stress unbalance) we have characterized the transition from creep, to slow earthquakes, to regular earthquakes (Scuderi et al., Nature Geoscience 2016). In our experimental configuration, the elastic response mentioned by Handwerger et al. is represented by the energy stored in the experimental apparatus. During the stick-slip seismic cycle (the lab analog of earthquakes), the contacts of the sliding surface, first are critically stressed and store elastic energy, that is then released during slip acceleration and failure. We did not develop a similar approach for Spriana since we think that fluid pressure plays a key-role and therefore it is the primary parameter to be tested in the lab.

Handwerger et al. (2016) say: “Catastrophic failure occurs only when the slip surface is characterized by rate-weakening friction and its lateral dimensions exceed a critical nucleation length h^* that is shorter

for higher effective stresses". Again, we totally agree with this and we have tested the role of critical nucleation length during pressurization of faults to better understand induced seismicity (Cappa et al., 2019 Science Advances). In this work we performed experiments similar to those performed in the present manuscript and used the lab derived RSF parameters to inform a 3D Hydromechanical model obeying RSF friction. Interestingly, we found that the model well fit the lab data and that an increase in fluid pressure would trigger accelerated creep similar to the one documented in the field and observed in the lab for Spriana. However, for this case too, we think that the accelerated creep can evolve into catastrophic behaviour if further weakening induced by fluid pressurization allows to overcome the second order R&S friction effect.

As the reviewer is saying, Helmstetter et al. (2004, Journal of Geophysical Research, vol. 109, B02409) have shown theoretically that creep acceleration of landslides could occur both in the velocity-weakening and velocity-strengthening regimes. We totally agree with this and we have laboratory data coupling frictional and fluid flow properties of the material supporting this statement. For example, for calcite fault gouge, during fluid pressure builds-up, the weakening induced by fluid pressure stimulation overcomes the slightly velocity strengthening behaviour of the material promoting the development of a dynamic instability (Scuderi et al., 2017, EPSL). In shales the weakening induced by fluids is counteracted by the strong velocity strengthening behaviour of the clay promoting accelerated slip not evolving into a dynamic instability (Scuderi and Collettini, 2018, JGR). The slip behaviour of the Spriana shear zone made of 40% of phyllosilicates seems to be in between calcite and shale, supporting the idea that the hydromechanical coupling strongly controls the slip behavior of a shear zone upon fluid pressurization.

General comment 2: The static friction coefficient is measured equal to 0.51 (line 111 and following). Some of the conclusions of the experimental data, applying the rate-and-state friction law implicitly assumes that the static friction coefficient is constant over the range of velocities 1-300 micron/s (whereas the dynamic friction coefficient will vary). If the strength of the landslide material in the experiment varies significantly, the interpretation of the author on the origin of the strengthening effect would be different. One way to visualise such effect would be to report the steady-state strength of the interface in the experiments for the range of velocities 1-300 micron/s and add this graph in supplementary material. Said differently, the authors have to ensure that the Mohr-Coulomb envelope of Fig. 2a is independent on the loading rate of the interface, in the range of loading velocities used in the experiments. For example, Thøgersen et al. (Geophysical Research Letters, 2019, doi: 10.1029/2019GL084436) have recently demonstrated that slow slip rates in faults are transient and controlled by the pre-stress on the interface and viscous/frictional dissipation (i.e. how the strength evolves with sliding velocity). This article has been published probably after the authors have submitted their work to Nature Communications, but the discussion there could be help the authors explain their results.

Reply: We apologize with the Reviewer but there may be a misunderstanding in the definitions. What we define as coefficient of friction at line 111 to be 0.51 is referred to the sliding friction of the material measured during steady state sliding at a velocity of 10 $\mu\text{m/s}$, more commonly called steady state friction (we have now specified it in the text). To fully answer the concerns of the Reviewer we point out that the data presented in Fig. 2a come from two different machines that have sheared material from the same sample at different velocities of 10 $\mu\text{m/s}$ (BRAVA) and 0.15 $\mu\text{m/s}$ (conventional Direct-Shear test, see Method), resulting in the same coefficient of friction. In general, the rate and state frictional properties represent second order variations in the coefficient of friction in the order of ~ 0.005 for the range of velocity from 1 to 300 $\mu\text{m/s}$. To answer the Reviewer's question, we have produced a figure that

we report below. In this figure, inset a) represents the evolution of the shear stress during a typical experiment that we have used to produce Figure 2a (main text), where the dashed line represents the averaged shear strength during steady state sliding at $v=10\mu\text{m/s}$, which is one of the point used to build the Coulomb-Mohr envelope in Figure 2a. Inset b) represents a zoom on the velocity step sequence where for each step we have averaged the shear stress at each velocity after steady state was achieved. In inset c), as asked by the Reviewer, we report on the original plot of Figure 2a (main text) the values we obtained from the analysis described above. Here, it is possible to observe that there is no significant deviation from the envelope that we reported originally. This is due to the fact that rate- and state-frictional properties represent second order variations in the coefficient of friction.

Dealing with the article by Thøgersen et al.; since: 1) we do observe only second order variations in the evolution of frictional strength within the tested sliding velocities (inset c below); 2) our data show that the slip behaviour is mainly controlled by the hydromechanical coupling during fluid pressure builds up

along the basal shear zone of the rockslide; and 3) the paper by Thøgersen et al. (2019) is a manuscript focussed on the debated relation (e.g. Gomberg et al., GRL 2016) between seismic moment and earthquake duration for regular vs. slow earthquakes, we have decided of not mentioning this manuscript.

General comment 3: An important outcome of the study, that would argue for publication in a high impact journal, would be to show predictions on the behavior of the Spriana rocks slide. For this, field data should be plotted in Fig. 4b and a discussion on what range of parameters or pore fluid pressure increase would induce a catastrophic failure of this landslide would be interesting. I suggest that the authors use their (high quality) data to discuss potential predictions or scenarios on how the rockslide could evolve in the future.

Reply: We thank the reviewer for this useful comment. Following his suggestion, we have plotted the rockslide data in former Fig. 4b (now Fig. 5e). Moreover, in the discussion we added some statements on the site conditions that could bring the rockslide to critical (slow to fast) conditions according to the observed match between our experimental results and *in situ* observations. The new text that support this is added at lines 265-273 (revised manuscript).

General comment to Fig. 1C: It is not clear how the different rockslide acceleration periods are defined. For example events 6 and 7 do not correspond to a significant increase of the deformation rate (the water-blue data are not correlated to the deformation-gray data). And what is the criterion to separate event 6 from event 7 ? Looking at the data, there is a high-water level period between events 4 and 5 and no clear acceleration of the landslide deformation. Any idea why?

In addition, I may suggest reporting the correlation coefficient between the water level and ground deformation and define events due to fluid increase pressure when the correlation coefficient is above a certain threshold.

Reply: The eight analysed acceleration periods were selected for comparison to laboratory data obtained from “pore pressure step” creep experiments. Each period reasonably meets the following requirements: a) significant short-term rise of groundwater level above the base level; b) increased water level maintained for a period before recession begins; c) impulsive acceleration (days to weeks) occurs and then displacement rates decrease to a “long-term” value (weeks to months). Actually, the selected periods correspond to significant increases of deformation rates, as suggested by the cumulative displacement curve in Fig. 1. Regarding the correlation between water level and deformation: there is no obvious correlation between pore pressure increase and rockslide response, and this is actually a major motivation of our study. As we now better specify in the main text, in practical applications it is very difficult to establish empirical (statistical) relationship between groundwater levels and large rockslide displacements/rates, hampering an “easy” prediction of their future behaviour. Our work shows that this complexity is related to the interplay between short-term undrained effects, modulating displacement rates, and long-term undrained to semi-drained effects of pore pressures causing shear zone weakening.

General comment to Line 69: remove «innovative» (several experimental results using this machine have been already published. In the present study, the material is new, not the experimental set-up).

Reply: We agree with the reviewer on the fact that the experimental set up has already produced several published results. Nonetheless, published results are in the field of fault mechanics, seismology and induced seismicity, while our experiments (not the machine) are very innovative in the field of landslide

science. We believe that keeping this term here can better underline the overall novelty of our study, also in agreement with the reviewer's suggestions.

General comment to Line 76: enucleated -> nucleated

Reply: Thank you. Done.

General comment to Line 191: The sentence “slip rates larger than the hydraulic conductivity” does not make sense. Even if slip rate and hydraulic conductivity have the same dimensions (m/s), they represent two different processes and their values cannot be compared to each others.

Reply: We understand the concern of the reviewer and we try to clarify this point here. Our intention is not to make a direct comparison between two velocities that intrinsically describe: one a physical property of the gouge (hydraulic conductivity) and another a spontaneous evolution in response to perturbations in the stress field (slip velocity). Following Figure 3 and Figure 4a (now Fig. 4), we observe that at slow creep rate the gouge shows an equilibrium in porosity because we do not observe any compaction or dilation, and coincidentally the hydraulic conductivity matches the creep velocity. The values of hydraulic conductivity that we report are measured via constant head permeability tests that are conducted under quasi-static conditions, resembling the conditions of the slow creep rate. The coupling of these observations implies that the values of hydraulic conductivity that we have measured can guarantee full saturation and fluid equilibration within the gouge at that porosity and shear velocity. As we perturb the system by means of fluid pressure, we observe acceleration and dilation. A sudden change in porosity (dilation) during slip acceleration is difficult to be immediately balanced by fluid diffusion favouring undrained fluid-to-solid hydro-mechanical coupling. Slow diffusion of the fluids during this stage may be responsible for an undrained response to rapid acceleration, then as time elapses and slip accumulates, fluid slowly diffuse (semi-drained response) but the shear zone slip at a faster rate due to a lowered effective stress.

We have modified the text in: “During the fluid pressure-step creep experiments, the system consistently behaves as described above when measured slip rates overcome a certain velocity threshold (Fig. 4). Above this threshold, the systematic observation of shear zone dilatancy corresponding to short-term pore pressure changes suggests the occurrence of direct, undrained fluid-to-solid hydro-mechanical coupling^{47,48}”.

General comment to Line 241: “shear strength of 0.89”, what is the unit (MPa) or is it normalized (like a friction coefficient)? It is in [MPa]??

Reply: Thank you, Reviewer 1 made exactly the same comment. There was typo: now we have corrected “shear strength of 0.89” to “shear strength of 0.89 τ_s ”

REVIEWERS' COMMENTS:

Reviewer #1 (Remarks to the Author):

Dear Editor,

Please find my evaluation of the manuscript entitled:

Slow to fast transition of giant creeping rockslides modulated by undrained loading in basal shear zones

By Agliardi et al.

The authors improved greatly the manuscript. I went throughout the text and the modifications.

They followed carefully my recommendations or in some cases justifies convincingly their options.

I think they also answer in a proper way to the other reviewers. As a consequence, I consider that the paper is ready for publication.

Sincerely yours

Reviewer #2 (Remarks to the Author):

Thanks to the authors for incorporating my suggestions into this revision. The revised manuscript now includes the relevant statistics for determining correlations and assessing similarities or differences in a supplementary table, and the description of those stats is succinct and clear in the main manuscript. As stated in my original review, I think this work is novel, well done, and should be of a broad interest to the geoscience community, and I would now recommend that the manuscript be accepted for publication. I'm satisfied with the revisions and don't need to see the manuscript again, but here are a couple more suggestions that I would ask the authors to consider as they finalize the manuscript:

34: Recommend defining "sub-critically stressed" in the manuscript itself. The authors did a clear job of explaining what definition they envision in the response to reviewers document, and I think it would be good, especially considering a broader audience, to include that definition here.

Fig 2a,b: Recommend adding error bars to points.

173: I don't see a clear demonstration of "gouge compaction that is directly proportional to the slip rate (Fig. 3a)" from that figure. It appears nonlinear after exceeding a threshold in Fig. 4.

Reviewer #3 (Remarks to the Author):

The authors did a great job to answer the criticisms I made and I consider the article could be published as it is.

Reply to final Comments from Reviewer #2

General comment: Thanks to the authors for incorporating my suggestions into this revision. The revised manuscript now includes the relevant statistics for determining correlations and assessing similarities or differences in a supplementary table, and the description of those stats is succinct and clear in the main manuscript. As stated in my original review, I think this work is novel, well done, and should be of a broad interest to the geoscience community, and I would now recommend that the manuscript be accepted for publication. I'm satisfied with the revisions and don't need to see the manuscript again, but here are a couple more suggestions that I would ask the authors to consider as they finalize the manuscript.

Reply: Thank you very much for your help and positive feedback, please find below the responses to your comments.

Comment to Line 34: Recommend defining "sub-critically stressed" in the manuscript itself. The authors did a clear job of explaining what definition they envision in the response to reviewers document, and I think it would be good, especially considering a broader audience, to include that definition here.

Reply: We added a very brief explanation of the term immediately after the term occurrence.

Comment to Fig. 2a,b: Recommend adding error bars to points.

Reply: In this case, error bars would be smaller than point symbols. We have highlighted this in the figure caption.

Comment to Line 173: I don't see a clear demonstration of "gouge compaction that is directly proportional to the slip rate (Fig. 3a)" from that figure. It appears nonlinear after exceeding a threshold in Fig. 4.

Reply: Thank you for the comment. We slightly modified the text to "Creep II is associated to gouge compaction that increases with increasing slip rate" and added a reference to Fig. 4, too. "(Figs. 3c and 4)".

Reply to final Comments from Reviewer #3

Comment: The authors did a great job to answer the criticisms I made and I consider the article could be published as it is.

Reply: Thank you very much for your help and very positive feedback.